# Separable dorsal raphe dopamine projections mimic the facets of a loneliness-like state

Christopher R Lee[1,2,3†], Gillian A Matthews[1,4†], Mackenzie E Lemieux[1,4†], Elizabeth M Wasserlein[4], Matilde Borio[1,4], Raymundo L Miranda[1,2,4], Laurel R Keyes[1,3], Gates P Schneider[1,3], Caroline Jia[1,2,3], Andrea Tran[1], Faith Aloboudi[1], May G Chan[1,3], Enzo Peroni[4], Grace Pereira[4], Alba López-Moraga[4], Anna Pallé[1,4], Eyal Y Kimchi[4], Nancy Padilla-Coreano[1,4], Romy Wichmann[1,3,4], Kay M Tye[1,2,3,4,5]*

[1]Salk Institute for Biological Studies, La Jolla, United States; [2]Neurosciences Graduate Program, University of California San Diego, La Jolla, United States; [3]Howard Hughes Medical Institute, La Jolla, United States; [4]The Picower Institute for Learning and Memory, Department of Brain and Cognitive Sciences, Massachusetts Institute of Technology, Cambridge, United States; [5]Kavli Institute for Brain and Mind, La Jolla, United States

*For correspondence:
tye@salk.edu

†These authors contributed equally to this work

Competing interest: The authors declare that no competing interests exist.

## eLife Assessment

This study dissects the function of 3 outputs of a specific population of modulatory neurons, dorsal raphe dopamine neurons, in social and affective behavior. It provides **valuable** information that both confirms prior results and provides new insights. The strength of the evidence is **convincing**, based on cutting-edge approaches and analysis. This study will be of interest to behavioral and systems neuroscientists, especially those interested in social and emotional behavior.

**Abstract** Affiliative social connections facilitate well-being and survival in numerous species. Engaging in social interactions requires positive or negative motivational drive, elicited through coordinated activity across neural circuits. However, the identity, interconnectivity, and functional encoding of social information within these circuits remains poorly understood. Here, we focus on downstream projections of dorsal raphe nucleus (DRN) dopamine neurons (DRN$^{DAT}$) in mice, which we previously implicated in social motivation alongside an aversive affective state. We show that three prominent DRN$^{DAT}$ projections – to the bed nucleus of the stria terminalis (BNST), central amygdala (CeA), and posterior basolateral amygdala (BLP) – play separable roles in behavior, despite substantial collateralization. Photoactivation of the DRN$^{DAT}$-CeA projection promoted social behavior and photostimulation of the DRN$^{DAT}$-BNST projection promoted exploratory behavior, while the DRN$^{DAT}$-BLP projection supported place avoidance, suggesting a negative affective state. Downstream regions showed diverse receptor expression, poising DRN$^{DAT}$ neurons to act through dopamine, neuropeptide, and glutamate transmission. Furthermore, we show ex vivo that the effect of DRN$^{DAT}$ photostimulation on downstream neuron excitability depended on region and baseline cell properties, resulting in excitatory responses in BNST cells and diverse responses in CeA and BLP. Finally, in vivo microendoscopic cellular-resolution recordings in the CeA with DRN$^{DAT}$ photo-stimulation revealed a correlation between social behavior and neurons excited by social stimuli, suggesting that increased dopamine tone may recruit different CeA neurons to social ensembles. Collectively, these circuit features may facilitate a coordinated, but flexible, response in the presence

of social stimuli that can be flexibly guided based on the internal social homeostatic need state of the individual.

## Introduction

A close social network confers a survival advantage, both in the wild and in the laboratory (*Yee et al., 2008*; *Koto et al., 2015*; *Silk et al., 2010*). Indeed, our brains have evolved to adapt to many changing conditions, including when we are with others and when we are alone. Many neuromodulatory systems and neural circuits engaged in social behaviors may serve a distinct function when social stimuli are not present. In non-social contexts, dopamine transporter-expressing dorsal raphe nucleus (DRN^DAT) neurons can promote incentive memory expression (*Lin et al., 2020*), antinociception (*Li et al., 2016*; *Meyer et al., 2009*; *Yu et al., 2021*), fear response (*Groessl et al., 2018*), and arousal (*Cho et al., 2017*; *Lu et al., 2006*; *Cho et al., 2021*) – showing a clear role in many functions essential for survival. Moreover, DRN^DAT neurons undergo synaptic strengthening after social isolation and increase responsiveness to social stimuli, and stimulation of these neurons induces a prosocial state (*Matthews et al., 2016*). Strikingly, a functional imaging study in humans similarly revealed that 10 hours of social isolation heightened midbrain responses to social stimuli (*Tomova et al., 2020*). In mice, we further demonstrated that photostimulation of DRN^DAT neurons not only promoted social preference, but also induced place avoidance, suggesting an aversive internal state (*Matthews et al., 2016*). This led us to infer a role for these neurons in motivating social approach, driven by the desire to quell a negative state (*Hull, 1943*), and playing a role in social homeostasis (*Lee et al., 2021*; *Matthews and Tye, 2019*).

Taken together, this suggests a broad functional role for DRN^DAT neurons in motivating adaptive, survival-promoting behaviors under both social and non-social conditions. While the multi-functional role of dopamine neurons in the DRN seems clear, it is yet unclear how these cells exert their influence at a circuit level, and the question remains: how do DRN^DAT neurons simultaneously motivate social approach while also inducing a negative state consistent with place avoidance? What downstream targets receive this signal, and how do they respond?

There are several circuit motifs and neural encoding strategies that could enable DRN^DAT neurons to simultaneously regulate these behavioral states and motivate adaptive responses. In a drive-state sequence model, if these DRN^DAT neurons were the control center in the social homeostat (*Lee et al., 2021*; *Matthews and Tye, 2019*), the unpleasant state of being isolated could then feed forward in a sequential chain to induce motivation to rectify this social deficit. However, in an effector state activation model, many parallel actions may be taken to address the challenge, and a pervasive behavioral state may be triggered by a neuromodulatory broadcast signal. In a parallel circuit model, distinct functional roles may be associated with projection-defined subpopulations in parallel (e.g. *Han et al., 2017*; *Kim et al., 2013*; *Kohl et al., 2018*; *Lammel et al., 2011*; *Namburi et al., 2015*; *Senn et al., 2014*; *Tye et al., 2011*), and neurons may simultaneously encode multiple types of information (i.e. exhibit 'mixed selectivity' *Rigotti et al., 2013*; *Tian et al., 2016*) or behavioral output may be governed by context- or state-dependency (e.g. *Krzywkowski et al., 2020*; *Kyriazi et al., 2018*; *Lemos et al., 2012*; *Seo et al., 2019*). Yet, the mechanisms through which DRN^DAT neurons exert their influence over social behavior have yet to be unraveled.

Here, we addressed the question of how DRN^DAT neurons modulate both sociability and valence by exploring the functional role and anatomical targets of distinct DRN^DAT projections in mice. We show that parallel DRN^DAT projections to different targets play separable roles in behavior, in spite of their heavily-collateralizing anatomical arrangement. Downstream, we find that within DRN^DAT terminal fields, there is spatial segregation of dopamine and neuropeptide receptor expression. Furthermore, photostimulation of DRN^DAT inputs can modulate downstream neuronal excitability depending on their baseline cell properties. Lastly, we find that DRN^DAT input enables a shift in central amygdala dynamics that allows it to predict social preference. These findings highlight the anatomical and functional heterogeneity that exists at multiple levels within the DRN^DAT system. We suggest this organization may underlie the capacity of the DRN^DAT system to exert a broad influence over different forms of behavior: allowing coordinated control over downstream neuronal activity and across the brain to signal a behavioral state that mimics a loneliness-like phenotype.

# Results

## DRN^DAT neurons project to and exhibit dense collateralization to distinct subregions of the amygdala and extended amygdala

To explore the circuit motifs (*Tye, 2018*) and computational implementation (*Lockwood et al., 2020*) through which the DRN^DAT system might operate, we examined whether discrete DRN^DAT projections underlie distinct features of behavior. Prominent DRN^DAT projections were identified by quantifying downstream fluorescence following Cre-dependent expression of eYFP in DAT^IREScre (B6.SJL-^Slc6a3tm1.1(cre) ^Bkmn/J) mice (*Matthews et al., 2016*; *Bäckman et al., 2006*; *Cardozo Pinto et al., 2019*; *Lammel et al., 2015*; *Figure 1—figure supplement 1*). We observed a distinct pattern of innervation arising from the ventral tegmental area (VTA)^DAT and DRN^DAT subpopulations (*Figure 1A–D*), with DRN^DAT projections most densely targeting the oval nucleus of the BNST (ovBNST) and lateral nucleus of the central amygdala (CeL). We also observed weaker, but significant, input to the posterior part of the basolateral amygdala (BLP), consistent with previous tracing studies (*Lin et al., 2020*; *Cardozo Pinto et al., 2019*; *Hasue and Shammah-Lagnado, 2002*; *Meloni et al., 2006*; *Oh et al., 2014*). Given that the extended amygdala and basolateral amygdala complex have been implicated in aversion- (*Davis et al., 2010*; *Goode and Maren, 2017*; *Janak and Tye, 2015*; *Lebow and Chen, 2016*) and reward-related processes (*Namburi et al., 2015*; *Douglass et al., 2017*; *Jennings et al., 2013*; *Kim et al., 2017*; *Tye et al., 2008*; *Tye et al., 2010*; *Bayless et al., 2023*), and connect with hindbrain motor nuclei to elicit autonomic and behavioral changes, we focused on these DRN^DAT projections (*Figure 1D*).

We next considered the anatomical organization of these projections to determine whether form gives rise to function. In other words, we investigated whether DRN^DAT outputs exhibit a circuit arrangement that facilitates a coordinated behavioral response. Axonal collateralization is one circuit feature that facilitates coordinated activity across broadly distributed structures (*Rockland, 2018*). Although VTA^DAT projections to striatal and cortical regions typically show little evidence of collateralization (*Aransay et al., 2015*; *Beier et al., 2015*; *Lerner et al., 2015*; *Matsuda et al., 2009*; *Moore and Bloom, 1978*), in contrast, DRN serotonergic neurons collateralize heavily to innervate the prefrontal cortex, striatum, midbrain, and amygdala (*Gagnon and Parent, 2014*; *van der Kooy and Hattori, 1980*; *Waselus et al., 2011*). However, it has yet to be determined whether DRN^DAT neurons are endowed with this property.

To assess whether DRN^DAT neurons exhibit axon collaterals, we performed dual retrograde tracing with fluorophore-conjugated cholera toxin subunit B (CTB; *Beyeler et al., 2018*). We injected each tracer into two of the three downstream sites (BNST, CeA, and/or BLP; *Figure 1E, F*, *Figure 1—figure supplement 2A–C*) and, after 7 days for retrograde transport, we examined CTB-expressing cells in the DRN that were co-labeled with tyrosine hydroxylase (TH; *Figure 1G*). CTB injections into the BNST and CeA resulted in numerous TH + cells labeled with both CTB-conjugated fluorophores, but fewer dual-labeled cells were observed when injections were placed in the BNST and BLP, or CeA and BLP (*Figure 1H*, *Figure 1—figure supplement 2D, E*). These data suggest significant collateralization to the extended amygdala, which includes the BNST and CeA (*Lin et al., 2020*; *Janak and Tye, 2015*). To confirm the presence of axon collaterals, we employed an intersectional viral strategy to selectively label CeA-projecting DRN^DAT neurons with cytoplasmic eYFP (*Figure 1—figure supplement 2F, G*). This resulted in eYFP-expressing terminals both in the CeA and in the BNST (*Figure 1—figure supplement 2H, I*).

## DRN^DAT-BLP photostimulation promotes place avoidance

We next considered whether DRN^DAT projections to the BNST, CeA, and BLP play separable or overlapping functional roles in modulating behavior. VTA dopaminergic input to the BNST and CeA has been implicated in threat discrimination (*De Bundel et al., 2016*; *Jo et al., 2018*), anxiety-related behavior (*de la Mora et al., 2012*), and drug-induced reward (*Eiler et al., 2003*; *Epping-Jordan et al., 1998*; *Rezayof et al., 2002*; *Thiel et al., 2010*), while in the BLA complex, dopamine signaling supports both fear (*Bissière et al., 2003*; *Fadok et al., 2009*; *Guarraci et al., 1999*; *de Oliveira et al., 2011*) and appetitive learning (*Tye et al., 2010*; *Lutas et al., 2019*). However, the question remains: do the same DRN^DAT projection neurons mediate different facets of a loneliness-like state, such as aversion, vigilance, and social motivation?

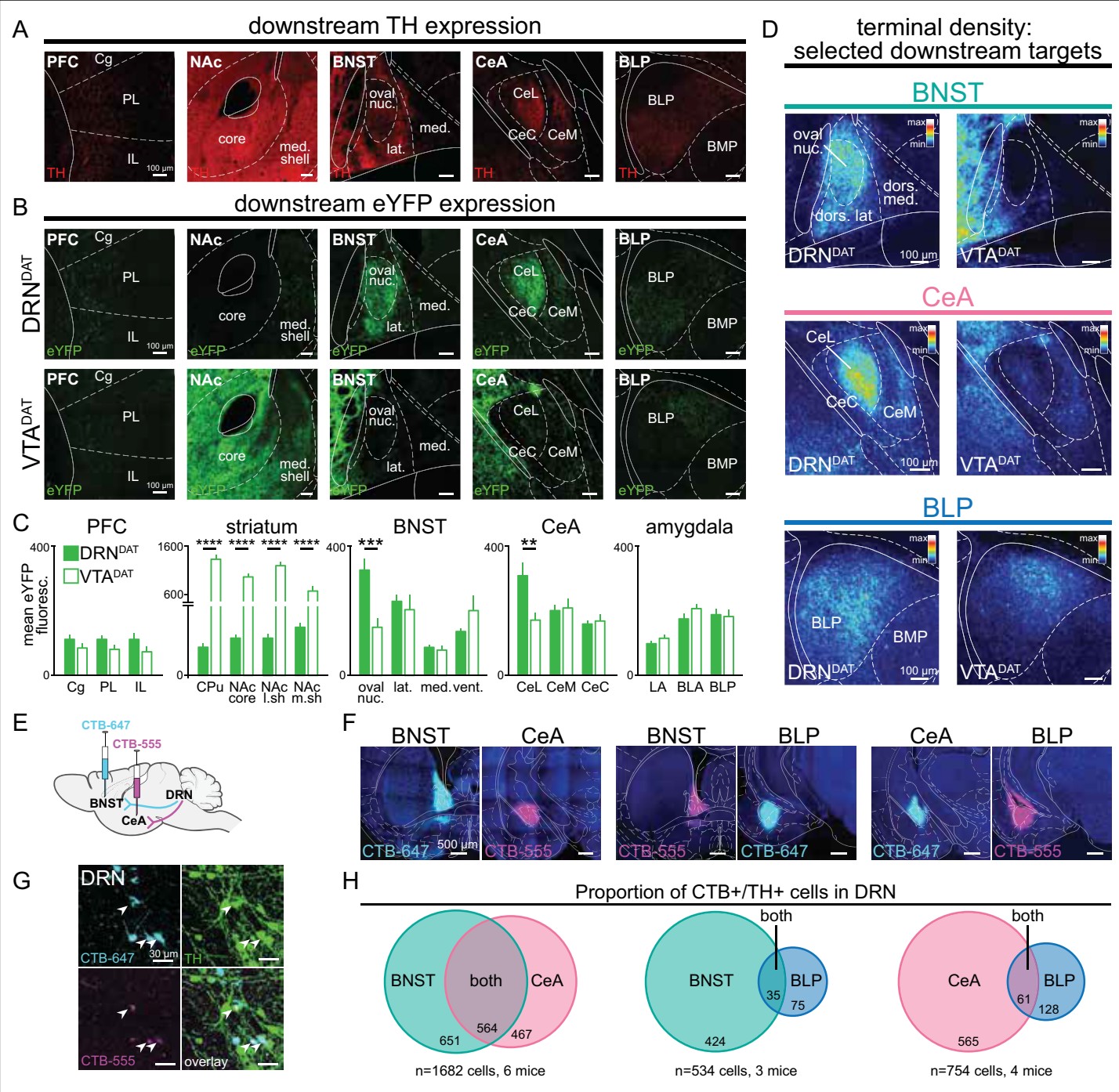

**Figure 1.** DRN[DAT] and VTA[DAT] afferents target distinct downstream regions. (**A**) Example images of downstream regions showing TH expression from immunohistochemistry. (**B**) eYFP expression in the prefrontal cortex (PFC), nucleus accumbens (NAc), bed nucleus of the stria terminalis (BNST), central amygdala (CeA), and posterior basolateral amygdala (BLP) following injection into the DRN (upper panels) and the VTA (lower panels). (**C**) Quantification of mean eYFP fluorescence in subregions from each structure (PFC: n=18 and 14 sections, striatum: n=20 and 21 sections, BNST: n=14 and 13 sections, CeA: n=24 and 27 sections, amygdala: n=45 and 51 sections from DRN and VTA injections, respectively, from 6 mice). eYFP fluorescence was significantly greater following VTA injection in all striatal subregions (unpaired t-test: CPu: $t_{39}$=13.23, p<0.0001; NAc core: $t_{39}$=13.56, p<0.0001; NAc lateral shell: $t_{31}$=13.01, p<0.0001; NAc medial shell: $t_{37}$=4.49, p<0.0001), and significantly greater following DRN injection in the BNST oval nucleus (unpaired t-test: $t_{22}$=3.95, p=0.0007) and CeA lateral division (unpaired t-test: $t_{34}$=3.18, p=0.0031). (**D**) Images from three selected downstream targets showing average terminal density in the middle anteroposterior (AP) region following eYFP expression in DRN[DAT] (left) or VTA[DAT] (right) neurons. (**E**) The retrograde tracer cholera toxin subunit-B (CTB) conjugated to Alexa Fluor 555 (CTB-555, pseudo-colored magenta) or Alexa-Fluor 647 (CTB-647, pseudo-colored cyan) was injected into two downstream targets. (**F**) Confocal images showing representative injection sites for dual BNST and

*Figure 1 continued on next page*

*Figure 1 continued*

CeA injections (left panels), BNST and BLP (center panels), and CeA and BLP (right panels). (**G**) High-magnification images of DRN cells expressing CTB-555 (magenta), CTB-647 (cyan), and TH (green) following injection into the BNST and CeA. White arrows indicate triple-labeled cells. (**H**) Venn diagrams showing the proportion of CTB+/TH + cells in the DRN following dual injections placed in the BNST and CeA (left), BNST and BLP (center), or CeA and BLP (right). When injections were placed in the BNST and CeA, dual CTB-labeled TH + cells constituted 46% of all BNST projectors and 55% of all CeA projectors. In contrast, when injections were placed in the BNST and BLP, or CeA and BLP, the proportion of dual-labeled cells was considerably lower (7.6% of BNST projectors and 9.7% of CeA projectors). Bar graphs show mean ± SEM. *p<0.05, **p<0.01, ***p<0.001, ****p<0.0001. PFC: Cg = cingulate cortex, PL = prelimbic cortex, IL = infralimbic cortex; striatum: CPu = caudate putamen, NAc core = nucleus accumbens core, NAc l.sh.=nucleus accumbens lateral shell, NAc m.sh.=nucleus accumbens medial shell; BNST: oval nuc.=BNST oval nucleus, lat.=BNST lateral division, med.=BNST medial division, vent.=BNST ventral part; CeL = central amygdala lateral division, CeM = central amygdala medial division, CeC = central amygdala capsular division; amygdala: LA = lateral amygdala, BLA = basolateral amygdala, BLP = basolateral amygdala posterior.

The online version of this article includes the following source data and figure supplement(s) for figure 1:

**Source data 1.** Mean DRN^DAT eYFP fluorescence in downstream regions, as shown in *Figure 1C*.

**Source data 2.** Colocalization counts of CTB+/TH + cells in the DRN, as shown in *Figure 1H*.

**Figure supplement 1.** DRN^DAT and VTA^DAT eYFP virus injection sites.

**Figure supplement 2.** Verification of dual-retrograde tracing strategy and intersectional approach to reveal axon collaterals.

**Figure supplement 2—source data 1.** Colocalization counts of CTB+/TH + cells in the DRN, as shown in *Figure 1—figure supplement 2C*.

To test the hypothesis that distinct DRN^DAT projections promote sociability, vigilance, and place avoidance (*Matthews et al., 2016*), we performed projection-specific ChR2-mediated photostimulation. We injected an AAV enabling Cre-dependent expression of ChR2 into the DRN of DAT::Cre male mice and implanted optic fibers over the BNST, CeA, or BLP (*Figure 2A*, *Figure 2—figure supplement 1A–F*). Given that we previously observed that behavioral effects of DRN^DAT photostimulation were predicted by an animal's social rank (*Matthews et al., 2016*), we also assessed relative social dominance using the tube test (*Lindzey et al., 1961*; *Wang et al., 2011*; *Zhou et al., 2018b*) prior to behavioral assays and photostimulation (*Figure 2A*, *Figure 2—figure supplement 1G, H*).

We first assessed whether photostimulation was sufficient to support place preference using the real-time place-preference (RTPP) assay. Here, we found that photostimulation of the DRN^DAT-BLP projection, but not the projection to the BNST or CeA, produced avoidance of the stimulation zone, relative to eYFP controls (*Figure 2B–G*). However, we did not find a significant correlation between social dominance and the magnitude of this effect (*Figure 2H–J*). Importantly, we did not detect an effect of photostimulation of DRN^DAT projections on operant intracranial self-stimulation (*Figure 2— figure supplement 2*).

## DRN^DAT-BNST photostimulation promotes non-social exploration

Next, we considered whether DRN^DAT projections to the BNST, CeA, or BLP play a role in increasing vigilance, a common behavioral marker in individuals experiencing loneliness (*Cacioppo et al., 2016*; *Cacioppo and Hawkley, 2009*). To assess how projection-specific photostimulation of DRN^DAT terminals affected exploratory behavior, we used the open field test (OFT) and elevated plus maze (EPM). While we found no effect of optical stimulation of DRN^DAT terminals on locomotion or time in center in the OFT (*Figure 3—figure supplement 1*), we found stimulation of DRN^DAT terminals in the BNST (but not in the CeA or BLP) resulted in a weak trend toward increased time spent in the open arm of the EPM (*Figure 3A–C*), which can be interpreted as exploratory behavior linked with a vigilant state (*Rodgers and Dalvi, 1997*). However, we found no correlation between social dominance and open arm time (*Figure 3D–F*). Strikingly, during social interaction with a novel juvenile in the home cage, we found that photoactivation of the DRN^DAT-BNST projection increased rearing behavior (a form of nonsocial exploration *Bailey and Crawley, 2009*; *Lever et al., 2006*; *Figure 3G–L*), an effect that was not previously observed with cell body photostimulation (*Matthews et al., 2016*). However, we did not find a significant correlation between social dominance and the expression of optically induced rearing behavior (*Figure 3J–L*).

## DRN^DAT-CeA photostimulation promotes sociability

To assess how projection-specific photostimulation of DRN^DAT terminals affected social preference, we used the three-chamber sociability task (*Moy et al., 2004*), where group-housed mice freely explored

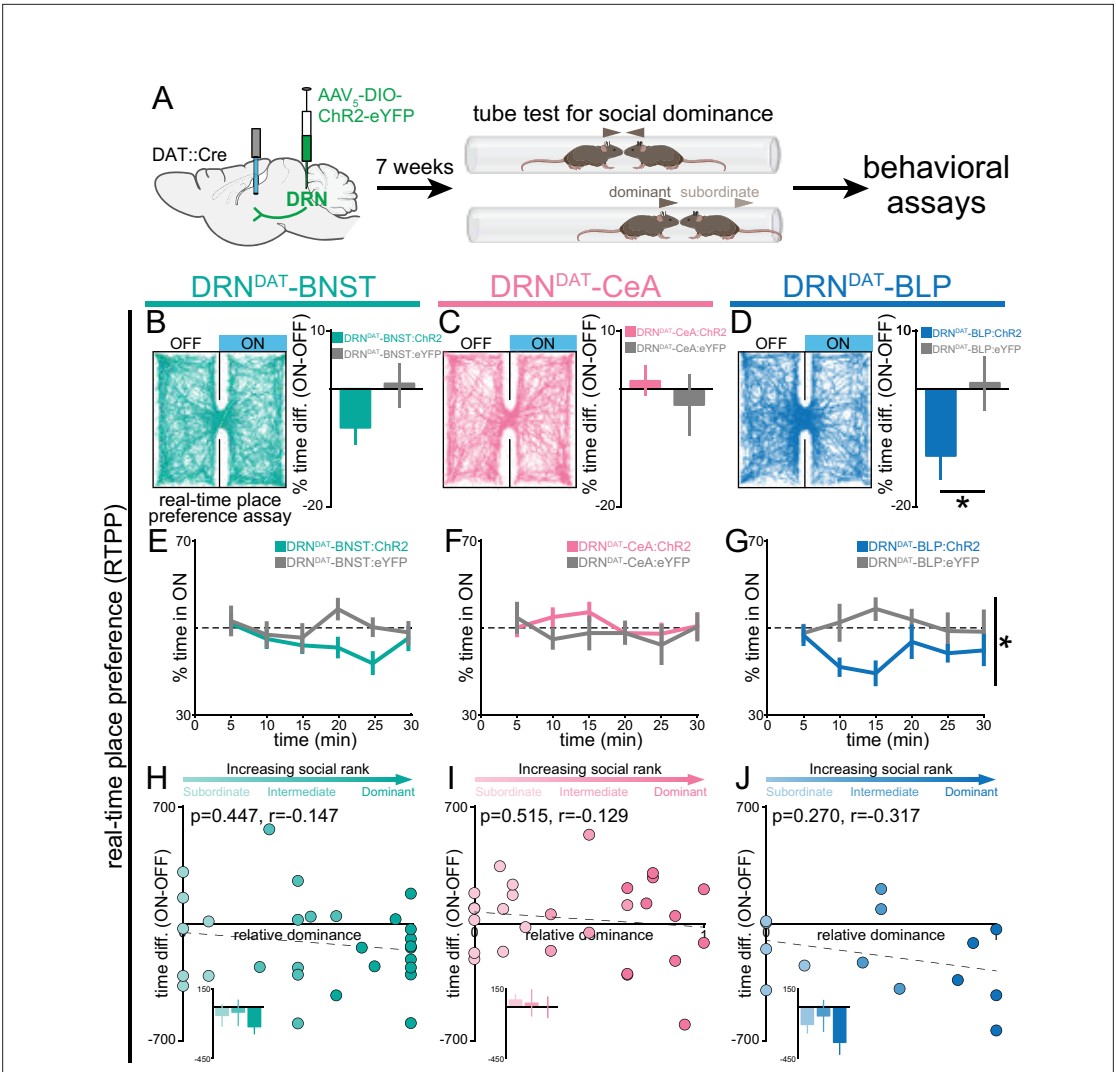

**Figure 2.** DRN^DAT-BLP (but not DRN^DAT-BNST or DRN^DAT-CeA) photostimulation promotes place avoidance. (**A**) AAV$_5$-DIO-ChR2-eYFP or AAV$_5$-DIO-eYFP was injected into the DRN of DAT::Cre mice and optic fibers implanted over the BNST, CeA, or BLP to photostimulate DRN^DAT terminals. After >7 weeks, viral expression cages of mice were assayed for social dominance using the tube test, prior to other behavioral tasks. (**B–D**) Left panels: example tracks of DRN^DAT-BNST:ChR2, DRN^DAT-CeA:ChR2, and DRN^DAT-BLP:ChR2 mice in the real-time place preference (RTPP) assay. Right panels: bar graphs showing the difference in % time spent in the stimulated ('ON') and unstimulated ('OFF') zones. There were no significant RTPP differences detected in (**B**) DRN^DAT-BNST:ChR2 (DRN^DAT-BNST:ChR2: N=29 mice, DRN^DAT-BNST:eYFP: N=14 mice; unpaired t-test: $t_{41}$=1.44, p=0.156) and (**C**) DRN^DAT-CeA:ChR2 mice (DRN^DAT-CeA:ChR2: N=28 mice, DRN^DAT-CeA:eYFP: N=13 mice; unpaired t-test: $t_{39}$=0.828, p=0.413) compared to their respective eYFP control mice groups. However, (**D**) DRN^DAT-BLP:ChR2 mice spent proportionally less time in the stimulated zone relative to DRN^DAT-BLP:eYFP mice (DRN^DAT-BLP:ChR2: N=14 mice, DRN^DAT-BLP:eYFP: N=8 mice; unpaired t-test: $t_{20}$=2.13, p=0.0455). (**E–G**) Time spent in the ON zone across the 30 min session. (**G**) DRN^DAT-BLP:ChR2 mice spent significantly less time in the ON zone relative to DRN^DAT-BLP:eYFP mice (DRN^DAT-BLP:ChR2: N=14 mice, DRN^DAT-BLP:eYFP: N=8 mice; repeated measures two-way ANOVA: $F_{1,20}$ = 4.53, main effect of opsin p=0.046). (**H–J**) Scatter plots showing relative dominance plotted against the difference in zone time (insets show mean values for subordinate, intermediate, and dominant mice) for (**H**) DRN^DAT-BNST, (**I**) DRN^DAT-CeA, or (**J**) DRN^DAT-BLP mice. Bar and line graphs display mean ± SEM. *p<0.05.

The online version of this article includes the following source data and figure supplement(s) for figure 2:

**Source data 1.** DRN^DAT-BNST:ChR2 RTPP percent time difference (ON-OFF), as shown in *Figure 2B*.

**Source data 2.** DRN^DAT-CeA:ChR2 RTPP percent time difference (ON-OFF), as shown in *Figure 2C*.

**Source data 3.** DRN^DAT-BLP:ChR2 RTPP percent time difference (ON-OFF), as shown in *Figure 2D*.

**Source data 4.** DRN^DAT-BNST:ChR2 RTPP percent time in ON (binned), as shown in *Figure 2E*.

**Source data 5.** DRN^DAT-CeA:ChR2 RTPP percent time in ON (binned), as shown in *Figure 2F*.

**Source data 6.** DRN^DAT-BLP:ChR2 RTPP percent time in ON (binned), as shown in *Figure 2G*.

*Figure 2 continued on next page*

*Figure 2 continued*

**Source data 7.** DRN$^{DAT}$-BNST:ChR2 RTPP percent time difference (ON-OFF) x relative dominance, as shown in *Figure 2H*.

**Source data 8.** DRN$^{DAT}$-CeA:ChR2 RTPP percent time difference (ON-OFF) x relative dominance, as shown in *Figure 2I*.

**Source data 9.** DRN$^{DAT}$-BLP:ChR2 RTPP percent time difference (ON-OFF) x relative dominance, as shown in *Figure 2J*.

**Figure supplement 1.** Fiber placement in DRN$^{DAT}$ downstream regions and stability of social dominance within cages.

**Figure supplement 1—source data 1.** Social rank stability, as shown in *Figure 2—figure supplement 1H*.

**Figure supplement 2.** Photostimulation of DRN$^{DAT}$ projections does not modify operant intra-cranial self-stimulation behavior.

**Figure supplement 2—source data 1.** DRN$^{DAT}$-BNST:ChR2 ICSS number of nose pokes, as shown in *Figure 2—figure supplement 2A*.

**Figure supplement 2—source data 2.** DRN$^{DAT}$-CeA:ChR2 ICSS number of nose pokes, as shown in *Figure 2—figure supplement 2B*.

**Figure supplement 2—source data 3.** DRN$^{DAT}$-BLP:ChR2 ICSS number of nose pokes, as shown in *Figure 2—figure supplement 2C*.

a chamber containing a novel juvenile mouse and a novel object at opposite ends (*Figure 4A–C*). This revealed that optical stimulation of the DRN$^{DAT}$-CeA projection increased social preference, but no significant effect was observed with photostimulation of either the DRN$^{DAT}$-BNST or DRN$^{DAT}$-BLP projections (*Figure 4D–F*; *Figure 4—figure supplement 1*). Furthermore, we found that the optically induced change in social preference in DRN$^{DAT}$-CeA mice was positively correlated with social dominance, suggesting that photostimulation elicited a greater increase in sociability in dominant mice (*Figure 4G–I*). This emulates the previous association found with photostimulation at the cell body level and social dominance (*Matthews et al., 2016*).

Next, to gain further insight into the functional divergence of DRN$^{DAT}$ projections in ethological behaviors, we assessed the effects of photostimulation on social interaction with a novel juvenile in the home cage. Here, photoactivation of the DRN$^{DAT}$-CeA projection modestly increased face sniffing of the juvenile mouse, consistent with a pro-social role for this projection (*Figure 4J–L*), although no correlation between optically induced change in face sniffing and social dominance was observed (*Figure 4M–O*). When we plotted the difference score (ON-OFF) for face sniffing against rearing (ON-OFF; *Figure 4—figure supplement 2A–C*), we observed that DRN$^{DAT}$-BNST mice tended to engage in more rearing and less face sniffing during photostimulation (i.e. located in the upper left quadrant), whereas DRN$^{DAT}$-CeA mice tended to exhibit less rearing and more face sniffing during photostimulation (i.e. located in the lower right quadrant).

To explore the relationship between social dominance and baseline behavioral profile, we applied a data-driven approach by examining behavioral measures obtained from different assays in a correlation matrix (*Figure 4—figure supplement 2D*). This showed a weak, negative correlation between social dominance and open arm time in the elevated plus maze (EPM) – consistent with a previous report of higher trait anxiety in dominant mice (*Larrieu et al., 2017*). However, social dominance did not correlate significantly with any other behavioral variable. Additionally, our analysis of baseline behavioral profile revealed a robust negative correlation between the time spent engaged in social sniffing and time spent rearing (*Figure 4—figure supplement 2D*). Furthermore, following dimensionality reduction on baseline behavioral variables, we did not find clearly differentiated clusters of high- and low-ranked mice (*Figure 4—figure supplement 2E*), suggesting that the variation governing these latent behavioral features is not related to social rank.

Finally, to determine whether DRN$^{DAT}$-CeA photostimulation affected the probability of behavioral state transition (*Füzesi et al., 2016*; *Lee et al., 2019*), we examined the sequential structure of behavior using a First-order Markov model (*Lee et al., 2019*; *Tejada et al., 2010*). Considering a two-state model consisting of 'social' and 'nonsocial' behaviors (*Figure 4P*), we found that photostimulation in DRN$^{DAT}$-CeA mice did not significantly change the probability of transitioning within social and nonsocial state (*Figure 4Q*), but did significantly change the probability of transitioning between social and nonsocial states (*Figure 4R*). This suggests that the DRN$^{DAT}$-CeA projection may increase engagement in social behavior by altering the overall structure of behavioral transitions.

## DRN$^{DAT}$ terminal fields contain spatially segregated dopamine and neuropeptide receptor populations

Our data suggest that DRN$^{DAT}$ projections exert divergent effects over behavior, despite substantial overlap in their upstream cells of origin. Given this overlap, we reasoned that one mechanism

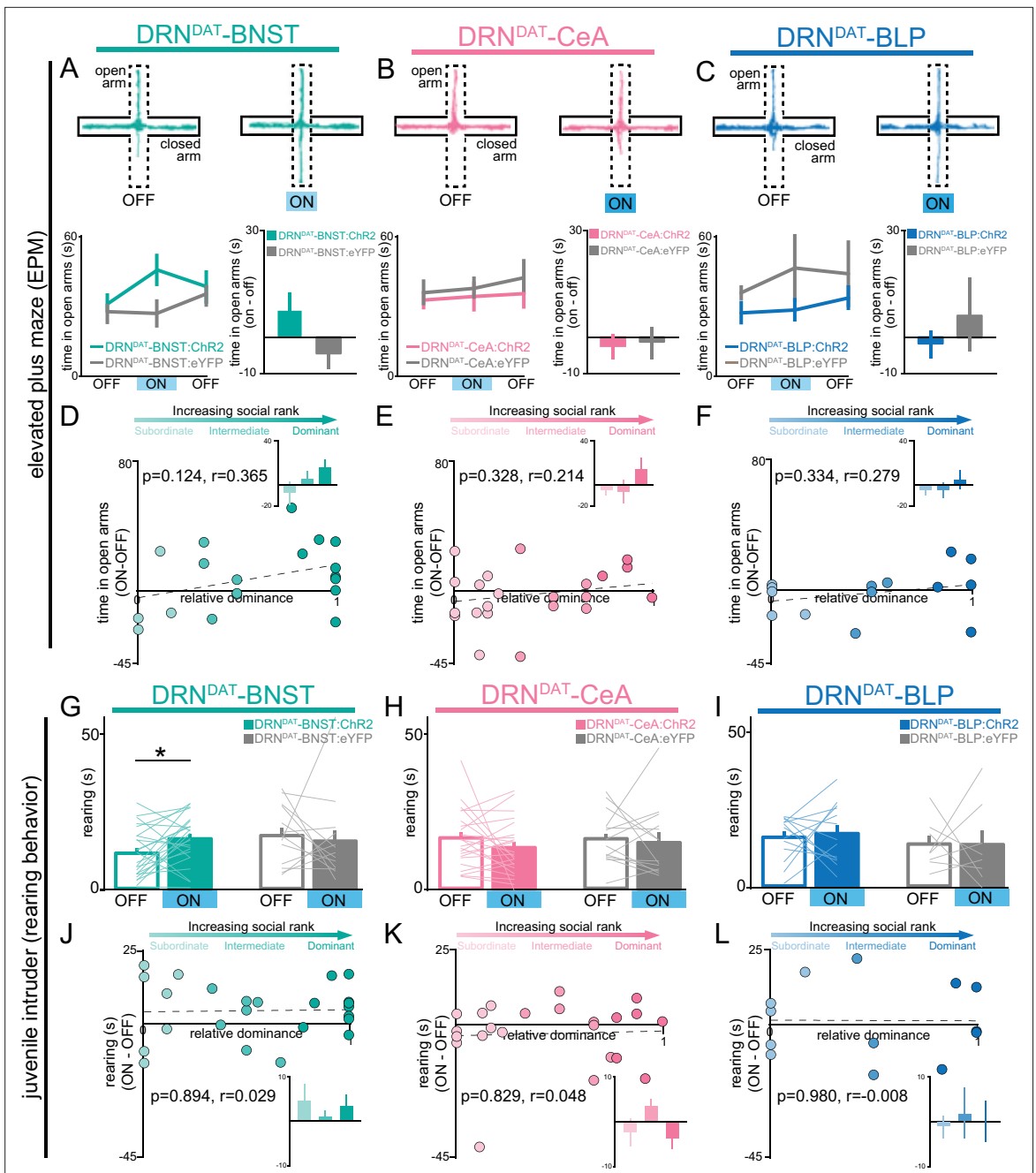

**Figure 3.** DRN[DAT]-BNST (but not DRN[DAT]-CeA or DRN[DAT]-BLP) photostimulation promotes non-social exploratory behavior. (**A–C**) Left panels: example tracks in the elevated plus maze (EPM) from a (**A**) DRN[DAT]-BNST:ChR2, (**B**), DRN[DAT]-CeA:ChR2, and (**C**), DRN[DAT]-BLP:ChR2 mouse. Upper right panels: time spent in the open arms of the EPM across the 15 min session. Photostimulation had no significant effect on time spent in the open arms of the EPM (two-way ANOVA, light x group interaction, BNST – $F_{2,50}$=2.008, p=0.145, CeA – $F_{2,72}$=0.118, p=0.889, BLP – $F_{2,40}$=0.354, p=0.704) for (**A**) DRN[DAT]-BNST, (**B**), DRN[DAT]-CeA, or (**C**) DRN[DAT]-BLP mice. Bottom right panels: difference in time spent in open arms of the EPM between the stimulation ON and first OFF epochs. Photostimulation had no significant effect on time spent in the open arms of the EPM for (**A**) DRN[DAT]-BNST (DRN[DAT]-BNST:ChR2: N=19 mice, DRN[DAT]-BNST:eYFP: N=10 mice; unpaired t-test: $t_{27}$=1.39, p=0.177), (**B**) DRN[DAT]-CeA (DRN[DAT]-CeA:ChR2: N=23 mice, DRN[DAT]-CeA:eYFP: N=14 mice; unpaired t-test: $t_{35}$=0.639, p=0.527), or (**C**) DRN[DAT]-BLP mice (DRN[DAT]-BLP:ChR2: N=14 mice, DRN[DAT]-BLP:eYFP: N=8 mice; unpaired t-test: $t_{20}$=0.759, p=0.457). (**D–F**) Scatter plots showing relative dominance plotted against the difference in the open arm zone time (insets show mean values for subordinate, intermediate, and dominant mice) for (**D**) DRN[DAT]-BNST, (**E**) DRN[DAT]-CeA, or (**F**) DRN[DAT]-BLP mice. (**G–I**) Home-cage behavior was assessed in the juvenile intruder assay across two counterbalanced sessions, one paired with photostimulation ('ON') and one without ('OFF') for (**G**) DRN[DAT]-BNST, (**H**) DRN[DAT]-CeA, or (**I**) DRN[DAT]-BLP mice. DRN[DAT]-BNST photostimulation increased time spent rearing (DRN[DAT]-BNST:ChR2: N=24 mice, DRN[DAT]-BNST:eYFP: N=13 mice; paired t-test: $t_{23}$=2.32, p=0.0298), but DRN[DAT]-CeA and DRN[DAT]-BLP photostimulation did not. (**J–L**) Scatter

*Figure 3 continued on next page*

*Figure 3 continued*

plots showing relative dominance plotted against the difference in rearing time with optical stimulation (ON-OFF) (insets show mean values for subordinate, intermediate, and dominant mice) for (**J**) DRN[DAT]-BNST, (**K**) DRN[DAT]-CeA, or (**L**) DRN[DAT]-BLP mice. Bar and line graphs display mean ± SEM. *p<0.05.

The online version of this article includes the following source data and figure supplement(s) for figure 3:

**Source data 1.** DRN[DAT]-BNST:ChR2 EPM open arm time (binned), as shown in *Figure 3A*.

**Source data 2.** DRN[DAT]-CeA:ChR2 EPM open arm time (binned), as shown in *Figure 3B*.

**Source data 3.** DRN[DAT]-BLP:ChR2 EPM open arm time (binned), as shown in *Figure 3C*.

**Source data 4.** DRN[DAT]-BNST:ChR2 EPM open arm time (ON-OFF) x relative dominance, as shown in *Figure 3D*.

**Source data 5.** DRN[DAT]-CeA:ChR2 EPM open arm time (ON-OFF) x relative dominance, as shown in *Figure 3E*.

**Source data 6.** DRN[DAT]-BLP:ChR2 EPM open arm time (ON-OFF) x relative dominance, as shown in *Figure 3F*.

**Source data 7.** DRN[DAT]-BNST:ChR2 juvenile intruder time spent rearing, as shown in *Figure 3G*.

**Source data 8.** DRN[DAT]-CeA:ChR2 juvenile intruder time spent rearing, as shown in *Figure 3H*.

**Source data 9.** DRN[DAT]-BLP:ChR2 juvenile intruder time spent rearing, as shown in *Figure 3I*.

**Source data 10.** DRN[DAT]-BNST:ChR2 juvenile intruder time spent rearing (ON-OFF) x relative dominance, as shown in *Figure 3J*.

**Source data 11.** DRN[DAT]-CeA:ChR2 juvenile intruder time spent rearing (ON-OFF) x relative dominance, as shown in *Figure 3K*.

**Source data 12.** DRN[DAT]-BLP:ChR2 juvenile intruder time spent rearing (ON-OFF) x relative dominance, as shown in *Figure 3L*.

**Figure supplement 1.** Photostimulation of DRN[DAT] projections does not modify locomotor or anxiety-like behavior.

**Figure supplement 1—source data 1.** DRN[DAT]-All projections:ChR2 OFT time in center, as shown in *Figure 3—figure supplement 1A–C*.

**Figure supplement 1—source data 2.** DRN[DAT]-All projections:ChR2 OFT distance traveled, as shown in *Figure 3—figure supplement 1A–C*.

through which these projections might achieve distinct behavioral effects is via differential recruitment of downstream signaling pathways. We, therefore, next considered whether the pattern of receptor expression differed within the DRN[DAT] terminal field of these downstream regions.

Subsets of DRN[DAT] neurons co-express vasoactive intestinal peptide (VIP) and neuropeptide-W (NPW; *Dougalis et al., 2012*; *Huang et al., 2019*; *Motoike et al., 2016*), and so we examined both dopamine (*Drd1* and *Drd2*) and neuropeptide (*Vipr2* and *Npbwr1*) receptor expression within DRN[DAT] terminal fields. To achieve this, we performed single molecule fluorescence in situ hybridization (smFISH) using RNAscope (*Figure 5—figure supplement 1A, B*). In the BNST and CeA, we observed a strikingly similar pattern of receptor expression with dense neuropeptide receptor expression in the oval BNST and ventromedial CeL, and a high degree of co-localization (*Figure 5A–H*, *Figure 5—figure supplement 1C–H*). In the BNST and CeA subregions containing the highest density of DRN[DAT] terminals, dopamine receptor expression was relatively more sparse, with *Drd2* more abundant than *Drd1*, as previously described (*Kim et al., 2017*; *De Bundel et al., 2016*; *de la Mora et al., 2012*; *McCullough et al., 2018a*; *McCullough et al., 2018b*; *Figure 5A–H*). The DRN[DAT] terminal field of the BLP displayed a markedly different receptor expression pattern, dominated by *Drd1* (*Figure 5I–L*, *Figure 5—figure supplement 1I–K*), consistent with previous reports (*de la Mora et al., 2012*; *Lutas et al., 2019*; *McCullough et al., 2018a*). Thus, in contrast to the BNST and CeA, the effects of DRN[DAT] input to the BLP may be predominantly mediated via D₁-receptor signaling. Collectively, this expression pattern suggests that the dopamine- and neuropeptide-mediated effects of DRN[DAT] input may be spatially segregated within downstream regions, providing the infrastructure for divergent modulation of cellular subsets.

## DRN[DAT] input has divergent effects on downstream cellular excitability

Our data suggest that DRN[DAT] projections exert divergent effects over behavior, despite substantial overlap in their upstream cells of origin. One mechanism through which these projections might achieve distinct behavioral effects is via differential modulation of activity in downstream neurons. The multi-transmitter phenotype of DRN[DAT] neurons (*Dougalis et al., 2012*; *Huang et al., 2019*; *Dougalis et al., 2017*; *Poulin et al., 2018*), regionally distinct downstream receptor expression, and the observed pre- and post-synaptic actions of exogenously applied dopamine (*Kash et al., 2008*; *Krawczyk et al., 2011*; *Kröner et al., 2005*; *Marowsky et al., 2005*; *Naylor et al., 2010*; *Rosenkranz and Grace, 1999*; *Rosenkranz and Grace, 2002*; *Silberman and Winder, 2013*) provide optimal

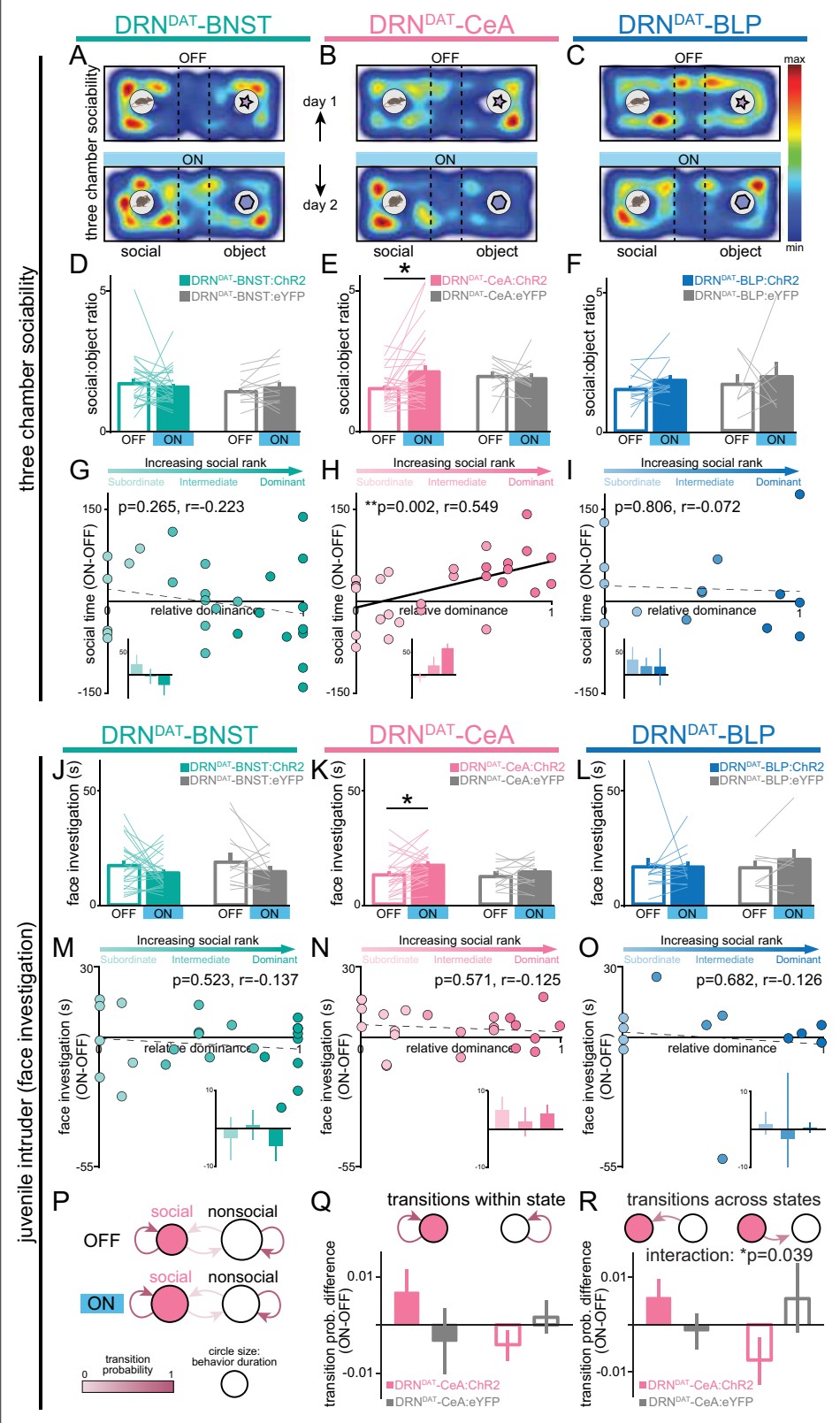

**Figure 4.** DRN[DAT]-CeA (but not DRN[DAT]-BNST or DRN[DAT]-BLP) photostimulation promotes sociability in a rank-dependent manner. (**A–C**) Heatmaps showing the relative location of ChR2-expressing mice in the three-chamber sociability assay, with optic fibers located over the (**A**) BNST, (**B**) CeA, or (**C**) BLP. The task was repeated across 2 days, with one session paired with photostimulation ('ON') and one without ('OFF'). (**D–F**) Bar graphs showing

*Figure 4 continued on next page*

*Figure 4 continued*

social preference in three-chamber sociability assay. (**D**) Photostimulation of DRN[DAT]-BNST terminals (8 pulses of 5ms pulse-width 473 nm light, delivered at 30 Hz every 5 s) in ChR2-expressing mice (DRN[DAT]-BNST:ChR2) had no significant effect on time spent in the social zone relative to the object zone (DRN[DAT]-BNST:ChR2: N=27 mice, DRN[DAT]-BNST:eYFP: N=14 mice; 'social:object ratio'; paired t-test: $t_{26}$=0.552, p=0.586), (**E**) but increased social:object ratio for DRN[DAT]-CeA:ChR2 mice (DRN[DAT]-CeA:ChR2: N=29 mice, DRN[DAT]-CeA:eYFP: N=13 mice; paired t-test: $t_{28}$=2.91; corrected for multiple comparisons: p=0.021) (**F**) and had no significant effect for DRN[DAT]-BLP:ChR2 mice (DRN[DAT]-BLP:ChR2: N=14 mice, DRN[DAT]-BLP:eYFP: N=7 mice; paired t-test: $t_{13}$=1.62, p=0.130). (**G–I**) Scatter plots displaying relative dominance plotted against the change in social zone time with optical stimulation (ON-OFF) for (**G**) DRN[DAT]-BNST, (**H**) DRN[DAT]-CeA, or (**I**) DRN[DAT]-BLP mice, showing significant positive correlation in DRN[DAT]-CeA:ChR2 mice (Pearson's correlation: r=0.549, p=0.002, N=29 mice). Inset bar graphs show mean values for subordinate, intermediate, and dominant mice. (**J–L**) Home-cage behavior was assessed in the juvenile intruder assay across two counterbalanced sessions, one paired with photostimulation ('ON') and one without ('OFF') for (**J**) DRN[DAT]-BNST, (**K**) DRN[DAT]-CeA, or (**L**) DRN[DAT]-BLP mice. DRN[DAT]-CeA photostimulation in ChR2-expressing mice increased time spent engaged in face investigation with the juvenile mouse (DRN[DAT]-CeA:ChR2: N=22 mice, DRN[DAT]-CeA:eYFP: N=14 mice; paired t-test: $t_{22}$=2.36, p=0.027). (**M–O**) Scatter plots showing relative dominance plotted against the difference in face investigation time with optical stimulation (ON-OFF) (insets show mean values for subordinate, intermediate, and dominant mice) for (**M**) DRN[DAT]-BNST, (**N**) DRN[DAT]-CeA, or (**O**) DRN[DAT]-BLP mice. (**P**) A two-state Markov model was used to examine behavioral transitions during the juvenile intruder assay for DRN[DAT]-CeA mice. (**Q, R**) Bar graphs showing the difference in transition probability (ON-OFF) for (**Q**) within-state transitions and (**R**) across-state transitions, for DRN[DAT]-CeA:ChR2 and DRN[DAT]-CeA:eYFP mice. There was no significant difference between ChR2 and eYFP groups for the change in within-state transition probability (DRN[DAT]-CeA:ChR2: N=22 mice, DRN[DAT]-CeA:eYFP: N=14 mice; two-way ANOVA: opsin x transition interaction, $F_{1,68}$=3.385, p=0.0702), (**R**) but there was a significant interaction between opsin and across-state transition probability (DRN[DAT]-CeA:ChR2: N=22 mice, DRN[DAT]-CeA:eYFP: N=14 mice; two-way ANOVA: opsin x transition interaction, $F_{1,68}$=4.452, p=0.0385) with photostimulation. Bar and line graphs display mean ± SEM. *p<0.05, **p<0.01.

The online version of this article includes the following source data and figure supplement(s) for figure 4:

**Source data 1.** DRN[DAT]-BNST:ChR2 three-chamber social:object ratio, as shown in *Figure 4D*.

**Source data 2.** DRN[DAT]-CeA:ChR2 three-chamber social:object ratio, as shown in *Figure 4E*.

**Source data 3.** DRN[DAT]-BLP:ChR2 three-chamber social:object ratio, as shown in *Figure 4F*.

**Source data 4.** DRN[DAT]-BNST:ChR2 time spent in social zone (ON-OFF) x relative dominance, as shown in *Figure 4G*.

**Source data 5.** DRN[DAT]-CeA:ChR2 time spent in social zone (ON-OFF) x relative dominance, as shown in *Figure 4H*.

**Source data 6.** DRN[DAT]-BLP:ChR2 time spent in social zone (ON-OFF) x relative dominance, as shown in *Figure 4I*.

**Source data 7.** DRN[DAT]-BNST:ChR2 juvenile intruder time spent in face investigation, as shown in *Figure 4J*.

**Source data 8.** DRN[DAT]-CeA:ChR2 juvenile intruder time spent in face investigation, as shown in *Figure 4K*.

**Source data 9.** DRN[DAT]-BLP:ChR2 juvenile intruder time spent in face investigation, as shown in *Figure 4L*.

**Source data 10.** DRN[DAT]-BNST:ChR2 juvenile intruder time spent in face investigation (ON-OFF) x relative dominance, as shown in *Figure 4M*.

**Source data 11.** DRN[DAT]-CeA:ChR2 juvenile intruder time spent in face investigation (ON-OFF) x relative dominance, as shown in *Figure 4N*.

**Source data 12.** DRN[DAT]-BLP:ChR2 juvenile intruder time spent in face investigation (ON-OFF) x relative dominance, as shown in *Figure 4O*.

**Source data 13.** DRN[DAT]-CeA:ChR2 juvenile intruder markov model (transition within state), as shown in *Figure 4Q*.

**Source data 14.** DRN[DAT]-CeA:ChR2 juvenile intruder markov model (transition across states), as shown in *Figure 4R*.

**Figure supplement 1.** Photostimulation of DRN[DAT] terminals in CeA (but not in BNST or BLP) increases time spent in three-chamber social zone.

**Figure supplement 1—source data 1.** DRN[DAT]-BNST:ChR2 three-chamber social zone time, as shown in *Figure 4—figure supplement 1A*.

**Figure supplement 1—source data 2.** DRN[DAT]-CeA:ChR2 three-chamber social zone time, as shown in *Figure 4—*

*Figure 4 continued*

**figure supplement 1B**.

**Figure supplement 1—source data 3.** DRN^DAT-BLP:ChR2 three-chamber social zone time, as shown in *Figure 4—figure supplement 1C*.

**Figure supplement 2.** Photostimulation of DRN^DAT projections effects on juvenile behavior, and analysis of baseline behavioral traits.

**Figure supplement 2—source data 1.** DRN^DAT-BNST:ChR2 juvenile intruder rearing time x face investigation time (ON-OFF), as shown in *Figure 4—figure supplement 2A*.

**Figure supplement 2—source data 2.** DRN^DAT-BNST:ChR2 juvenile intruder rearing time x face investigation time (ON-OFF), as shown in *Figure 4—figure supplement 2B*.

**Figure supplement 2—source data 3.** DRN^DAT-BNST:ChR2 juvenile intruder rearing time x face investigation time (ON-OFF), as shown in *Figure 4—figure supplement 2C*.

**Figure supplement 2—source data 4.** Baseline behavioral measures correlation matrix (r-values), as shown in *Figure 4—figure supplement 2D*.

**Figure supplement 2—source data 5.** Baseline behavioral measures correlation matrix (p-values), as shown in *Figure 4—figure supplement 2D*.

**Figure supplement 2—source data 6.** Baseline behavioral measures (raw values), as shown in *Figure 4—figure supplement 2D, E*.

---

conditions for diverse modulation of neural activity. However, it remains unknown how temporally precise activation of DRN^DAT terminals influences excitability at the single-cell level.

We, therefore, next examined how DRN^DAT input affects downstream excitability. To achieve this, we expressed ChR2 in DRN^DAT neurons and used ex vivo electrophysiology to record from downstream neurons (*Figure 6A–C*, *Figure 6—figure supplement 1A–C*). Optical stimulation at the resting membrane potential evoked both excitatory and inhibitory post-synaptic potentials (EPSPs and IPSPs) in downstream cells (*Figure 6D–F*), which were typically monosynaptic (*Figure 6—figure supplement 1D, E*). During spontaneous firing, BNST cells were universally excited, whereas more diverse responses were observed with the BLP and CeA (*Figure 6G–K*, *Figure 6—figure supplement 1F–G*). The fast rise and decay kinetics of the EPSP suggest an AMPAR-mediated potential, resulting from glutamate co-release (*Li et al., 2016*; *Matthews et al., 2016*), whereas the slow IPSP kinetics are consistent with opening of GIRK channels, which can occur via $D_2$-receptor (*Beckstead et al., 2004*; *Marcott et al., 2018*) or GABA-$_B$ receptor signaling (*Bettler et al., 2004*; *Destexhe and Sejnowski, 1995*; *Mackay et al., 2019*).

Given the diversity of responses observed in the CeA and BLP, we next examined these downstream cells in more detail. To assess whether baseline electrophysiological properties predicted the optically evoked response, we used unsupervised agglomerative hierarchical clustering to classify downstream cells (*Figure 6L, M*). This established approach has been successfully applied to electrophysiological datasets to reveal distinct neuronal subclasses (*Cauli et al., 2000*; *Guthman et al., 2020*; *Hou et al., 2016*). The resulting dendrograms yielded two major clusters in the CeA and BLP, with distinct electrophysiological characteristics (*Figure 6N–Q*, *Figure 6—figure supplement 1H–K*). CeA cells in cluster 1 represented 'late-firing' neurons, whereas cluster 2 was typical of 'regular-firing' neurons (*Chieng et al., 2006*; *Dumont et al., 2002*; *Lopez de Armentia and Sah, 2004*). Strikingly, these clusters exhibited dramatically different responses to DRN^DAT photostimulation, with cluster 1 'late-firing' neurons excited and cluster 2 'regular-firing' neurons mostly inhibited (*Figure 6O*). Similarly, BLP cells delineated into two major clusters, with properties characteristic of pyramidal neurons (cluster 1) and GABAergic interneurons (cluster 2; *Figure 6P, Q*). These clusters showed remarkably different responses to DRN^DAT input, with 93% of putative pyramidal neurons showing an inhibitory response, and 62% of putative GABAergic interneurons showing an excitatory response (*Figure 6Q*). In addition, clustering CeA and BLP cells together yielded a very similar result (*Figure 6—figure supplement 1L–N*). Thus, while photoactivation of DRN^DAT terminals elicits heterogeneous responses in downstream neurons, baseline cell properties strongly predict their response, suggesting robust synaptic organization. The opposing nature of these responses, in different neuronal subsets, suggests that – rather than inducing an overall augmentation or suppression of activity – DRN^DAT input may adjust the *pattern* of downstream activity, in order to exert a functional shift in behavior.

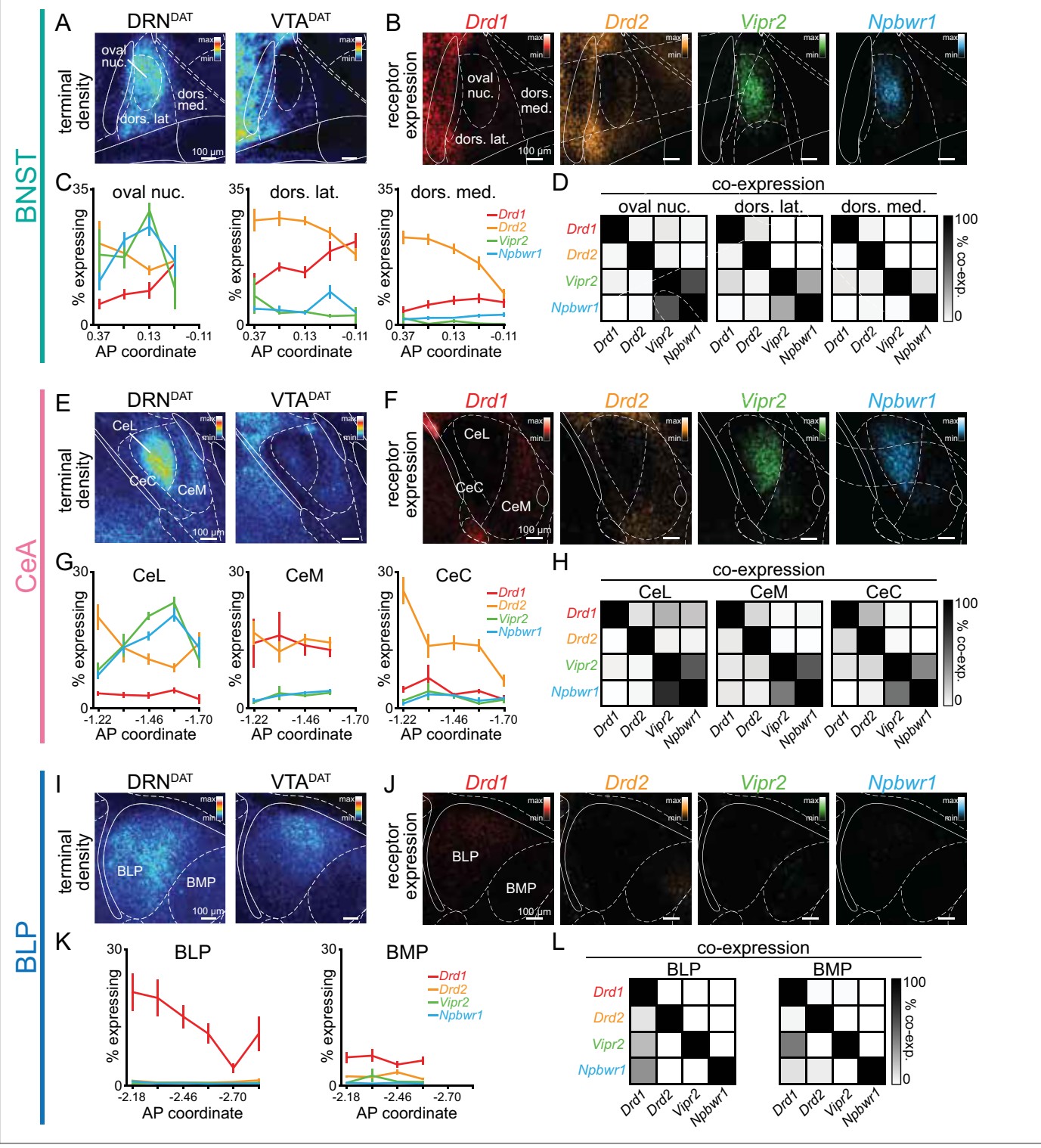

**Figure 5.** Spatial segregation of dopamine and neuropeptide receptor populations within DRN[DAT] terminal fields. (**A**) Mean projection of terminal density in the middle anteroposterior (AP) region of the BNST, following eYFP expression in DRN[DAT] (left) or VTA[DAT] (right) neurons. (**B**) Mean projection showing fluorescent puncta in the BNST indicating detection of *Drd1* (red), *Drd2* (yellow), *Vipr2* (green), or *Npbwr1* (blue) mRNA transcripts. (**C**) Line graphs showing the percent of cells expressing each receptor (≥5 puncta) across AP locations for the oval nucleus, dorsolateral BNST, and dorsomedial BNST (two-way ANOVA, oval nucleus: probe x AP interaction, $F_{9,160}=6.194$, $p<0.0001$, dorsolateral BNST: probe x AP interaction, $F_{12,167}=3.410$, $p=0.0002$, dorsomedial BNST: probe x AP interaction, $F_{12,161}=2.268$, $p=0.0110$). *Drd1*: $n=51,55,53$ *Drd2*: $n=52,55,53$ *Vipr2*: $n=37,39,37$ *Npbwr1*: $n=36,38,38$ sections, for oval nucleus, dorsolateral BNST, and dorsomedial BNST, respectively, from 4 mice. (**D**) Matrices indicating overlap between mRNA-expressing cells:

*Figure 5 continued on next page*

*Figure 5 continued*

square shade indicates the percent of cells expressing the gene in the column from within cells expressing the gene in the row. (**E**) Mean projection of terminal density in the middle AP region of the CeA, following eYFP expression in DRN$^{DAT}$ (left) or VTA$^{DAT}$ (right) neurons. (**F**) Mean projection showing fluorescent puncta in the CeA indicating mRNA expression.(**G**) Line graphs showing the % of cells expressing each receptor (≥5 puncta) across AP locations for the CeL, CeM, and CeC (two-way ANOVA, CeL: probe x AP interaction, $F_{12,220}$=8.664, p<0.0001, CeM: main effect of probe, $F_{3,186}$=60.30, p<0.0001, CeC: probe x AP interaction, $F_{12,218}$=4.883, p<0.0001). *Drd1: n*=47,40,47 *Drd2: n*=70,55,70 *Vipr2: n*=65,57,63 *Npbwr1: n*=62,50,60 sections, for CeL, CeM, and CeC, respectively, from 4 mice. (**H**) Matrices indicating overlap between mRNA-expressing cells. (**I**) Mean projection of terminal density in the middle AP region of the BLP, following eYFP expression in DRN$^{DAT}$ (left) or VTA$^{DAT}$ (right) neurons. (**J**) Mean projection showing fluorescent puncta in the BLP indicating mRNA expression. (**K**) Line graphs showing the percent of cells expressing each receptor (≥5 puncta) across AP locations for the BLP and BMP (two-way ANOVA, BLP: probe x AP interaction, $F_{15,176}$=2.165, p=0.0091, BMP: main effect of probe, $F_{3,141}$=56.92, p<0.0001). *Drd1: n*=55,44 *Drd2: n*=59,46 *Vipr2: n*=41,33 *Npbwr1: n*=45,34 sections, for BLP and BMP, respectively, from 4 mice. (**L**) Matrices indicating overlap between mRNA-expressing cells. Line graphs show mean ± SEM.

The online version of this article includes the following source data and figure supplement(s) for figure 5:

**Source data 1.** BNST RNAScope sub-regional probe expression (percent), as shown in *Figure 5C*.

**Source data 2.** BNST RNAScope sub-regional probe co-expression, as shown in *Figure 5D*.

**Source data 3.** CeA RNAScope sub-regional probe expression (percent), as shown in *Figure 5G*.

**Source data 4.** CeA RNAScope sub-regional probe co-expression, as shown in *Figure 5H*.

**Source data 5.** BLA RNAScope sub-regional probe expression (percent), as shown in *Figure 5K*.

**Source data 6.** BLA RNAScope sub-regional probe co-expression, as shown in *Figure 5L*.

**Figure supplement 1.** Analysis of mRNA expression using different thresholds qualitatively shows similar spatial pattern of dopamine and neuropeptide receptor expression in downstream regions.

**Figure supplement 1—source data 1.** Number of puncta x pixels occupied for all RNAScope probes, as shown in *Figure 5—figure supplement 1B*.

**Figure supplement 1—source data 2.** BNST RNAScope sub-regional probe expression (percent, threshold = 1 punctum/cell), as shown in *Figure 5—figure supplement 1D*.

**Figure supplement 1—source data 3.** BNST RNAScope sub-regional probe expression (percent, threshold = 3 puncta/cell), as shown in *Figure 5—figure supplement 1E*.

**Figure supplement 1—source data 4.** CeA RNAScope sub-regional probe expression (percent, threshold = 1 punctum/cell), as shown in *Figure 5—figure supplement 1G*.

**Figure supplement 1—source data 5.** CeA RNAScope sub-regional probe expression (percent, threshold = 3 puncta/cell), as shown in *Figure 5—figure supplement 1H*.

**Figure supplement 1—source data 6.** BLA RNAScope sub-regional probe expression (percent, threshold = 1 punctum/cell), as shown in *Figure 5—figure supplement 1J*.

**Figure supplement 1—source data 7.** BLA RNAScope sub-regional probe expression (percent, threshold = 3 puncta/cell), as shown in *Figure 5—figure supplement 1K*.

## DRN$^{DAT}$ input enables a functional shift in CeA dynamics to predict social preference

Our data thus far suggest that photostimulation of DRN$^{DAT}$ projections to downstream extended amygdala targets elicits divergent behaviors that are, together, congruent with a loneliness-like state, with the DRN$^{DAT}$-CeA projection promoting sociability. Considering the diversity of responses in the CeA elicited by DRN$^{DAT}$ input ex vivo, we next wondered how DRN$^{DAT}$ input into the CeA in vivo during a behaviorally relevant task may modify how the CeA represents social information. Neuromodulatory input has been previously shown to alter responses to salient stimuli—for instance, stimulation of VTA dopamine terminals increases the signal-to-noise ratio to aversive stimuli in projection-specific populations of prefrontal cortex neurons (*Vander Weele et al., 2018*). However, how DRN$^{DAT}$ input modifies the coding scheme of CeA neurons for social information remains unknown.

Therefore, to test the hypothesis that DRN$^{DAT}$ input alters the responses of CeA neurons to functionally relevant stimuli, we examined the dynamics of CeA neurons while simultaneously stimulating DRN$^{DAT}$ terminals during a three-chamber sociability task. To achieve this, we expressed the calcium indicator GCaMP7f nonspecifically in the CeA and either the red-shifted opsin ChrimsonR or a control fluorophore (TdTomato) in the DRN of DAT::Cre mice, and additionally implanted a gradient index (GRIN) lens over the CeA (*Figure 7A*, *Figure 7—figure supplement 1A*). This allowed us to stimulate DRN$^{DAT}$ terminals in the CeA while resolving single-cell calcium dynamics in the CeA in vivo

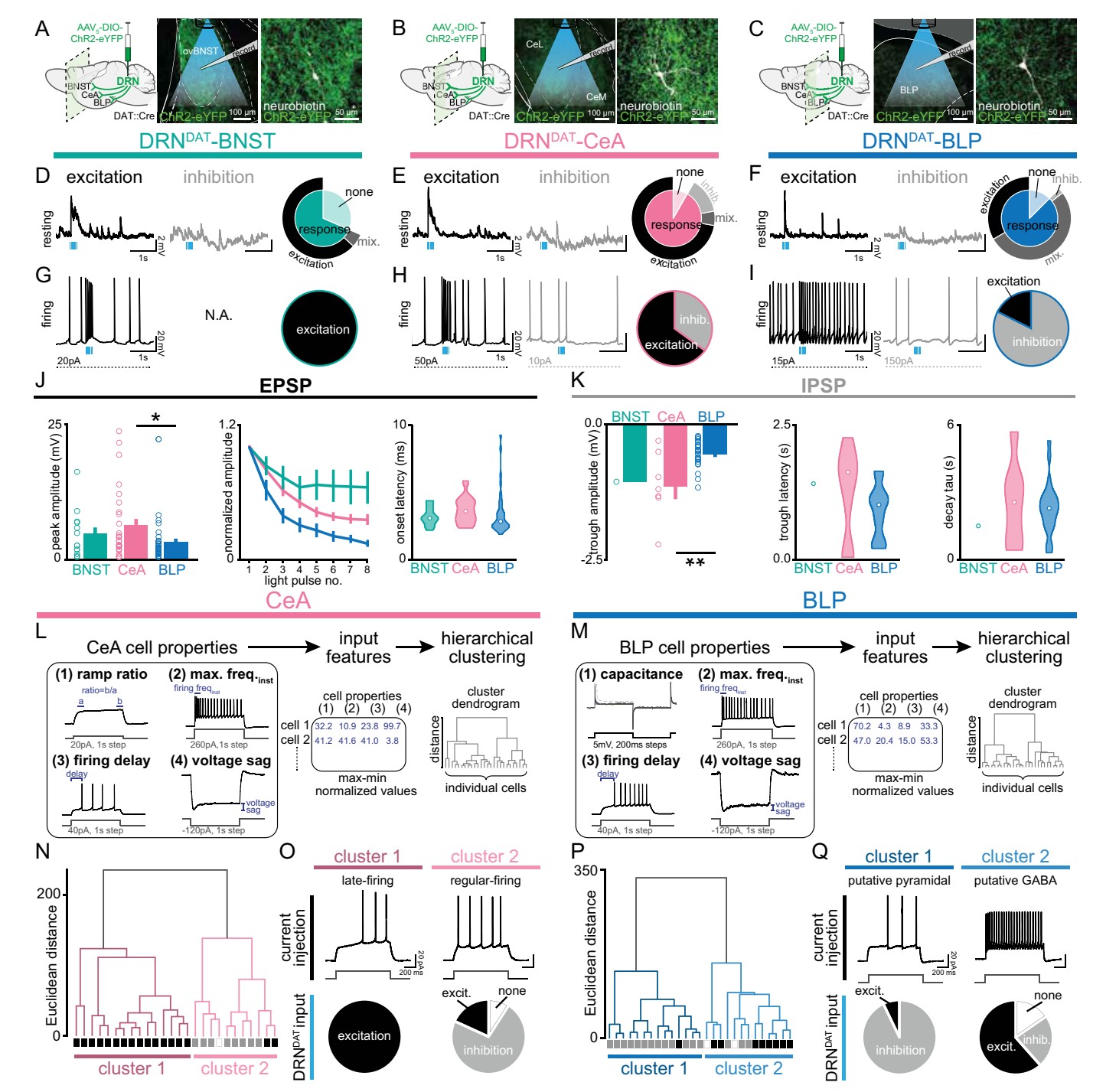

**Figure 6.** DRN$^{DAT}$ input distinctly influences downstream activity in each downstream target. (**A–C**) In mice expressing ChR2 in DRN$^{DAT}$ neurons, ex vivo electrophysiological recordings were made from (**A**) the BNST, (**B**) CeA, and (**C**) BLP. (**D–F**) Photostimulation of DRN$^{DAT}$ terminals with blue light (8 pulses delivered at 30 Hz) evoked both excitatory and inhibitory responses at resting membrane potentials in (**D**) the BNST, (**E**) CeA, and (**F**) BLP. Traces show single sweeps and pie charts indicate proportion of cells with no response ('none'), an EPSP only ('excitation'), an IPSP only ('inhibition'), or a mixed combination of EPSPs and IPSPs ('mix'). Recorded cells: BNST n=19, CeA n=36, BLP n=48. (**G–I**) When constant current was injected to elicit spontaneous firing, (**G**) BNST cells responded to photostimulation with an increase in firing ('excitation'), while (**H**) CeA and (**I**) BLP cells responded with an increase or a decrease in firing ('inhibition'). Recorded cells: BNST n=5, CeA n=20, BLP n=17. (**J**) Properties of the optically evoked excitatory post-synaptic potential (EPSP) at resting membrane potentials – left: peak amplitude (Kruskal-Wallis statistic = 6.790, p=0.0335; Dunn's posts-hoc tests: CeA vs BLP p=0.0378; middle: change in amplitude across light pulses; right: violin plots showing distribution of onset latencies (white circle indicates median).(**K**) Properties of the optically evoked inhibitory post-synaptic potential (IPSP) at resting membrane potentials – left panel: trough amplitude

*Figure 6 continued on next page*

*Figure 6 continued*

(one-way ANOVA, $F_{2,31}$=8.150, p=0.0014, CeA vs BLP: **p=0.0014); middle panel: violin plot showing latency to trough peak; right panel: violin plot showing tau for the current decay (white circle indicates median). (**L**) Workflow for agglomerative hierarchical clustering of CeA neurons and (**M**) BLP neurons. Four baseline electrical properties were used as input features (following max-min normalization) and Ward's method was used to generate a cluster dendrogram, grouping cells based on Euclidean distance. (**N**) Dendrogram for CeA cells indicating two major clusters, with their response to DRN$^{DAT}$ input indicated below each branch (excitatio*n* = black; inhibitio*n* = grey; no response = open). (**O**) Upper panels: cluster 1 showed baseline properties typical of 'late-firing' neurons and cluster 2 showed baseline properties typical of 'regular-firing' neurons. Lower panels: pie charts showing the response of cells in each cluster to DRN$^{DAT}$ input. (**P**) Dendrogram for BLP cells indicating two major clusters, with their response to DRN$^{DAT}$ input indicated below each branch (excitatio*n* = black; inhibitio*n* = grey; no response = open). (**Q**) Upper panels: cluster 1 showed baseline properties typical of pyramidal neurons and cluster 2 showed baseline properties typical of GABA interneurons. Lower panels: pie charts showing the response of cells in each cluster to DRN$^{DAT}$ input. Bar and line graphs show mean ± SEM. *p<0.05, **p<0.01.

The online version of this article includes the following source data and figure supplement(s) for figure 6:

**Source data 1.** BNST (resting) ex vivo responses to DRN$^{DAT}$ optical stimulation, as shown in *Figure 6D*.

**Source data 2.** CeA (resting) ex vivo responses to DRN$^{DAT}$ optical stimulation, as shown in *Figure 6E*.

**Source data 3.** BLP (resting) ex vivo responses to DRN$^{DAT}$ optical stimulation, as shown in *Figure 6F*.

**Source data 4.** BNST (firing) ex vivo responses to DRN$^{DAT}$ optical stimulation, as shown in *Figure 6G*.

**Source data 5.** CeA (firing) ex vivo responses to DRN$^{DAT}$ optical stimulation, as shown in *Figure 6H*.

**Source data 6.** BLP (firing) ex vivo responses to DRN$^{DAT}$ optical stimulation, as shown in *Figure 6I*.

**Source data 7.** BNST/CeA/BLP ex vivo EPSP peak amplitude in response to DRN$^{DAT}$ optical stimulation, as shown in *Figure 6J*.

**Source data 8.** BNST/CeA/BLP ex vivo EPSP normalized amplitude in response to DRN$^{DAT}$ optical stimulation, as shown in *Figure 6J*.

**Source data 9.** BNST/CeA/BLP ex vivo EPSP onset latency in response to DRN$^{DAT}$ optical stimulation, as shown in *Figure 6J*.

**Source data 10.** BNST/CeA/BLP ex vivo IPSP trough amplitude in response to DRN$^{DAT}$ optical stimulation, as shown in *Figure 6K*.

**Source data 11.** BNST/CeA/BLP ex vivo IPSP trough latency in response to DRN$^{DAT}$ optical stimulation, as shown in *Figure 6K*.

**Source data 12.** BNST/CeA/BLP ex vivo IPSP decay tau in response to DRN$^{DAT}$ optical stimulation, as shown in *Figure 6K*.

**Source data 13.** CeA ex vivo baseline cell properties used for hierarchical clustering, as shown in *Figure 6L–O*.

**Source data 14.** BLP ex vivo baseline cell properties used for hierarchical clustering, as shown in *Figure 6M–Q*.

**Figure supplement 1.** Effect of DRN$^{DAT}$ photostimulation on downstream cellular excitability ex vivo.

**Figure supplement 1—source data 1.** BNST/CeA/BLP ex vivo EPSP/IPSP normalized peak amplitude in response to DRN$^{DAT}$ optical stimulation with TTX/4AP application, as shown in *Figure 6—figure supplement 1E*.

**Figure supplement 1—source data 2.** BNST/CeA/BLP ex vivo EPSP/IPSP peak/trough pre-stimulation membrane potential, as shown in *Figure 6—figure supplement 1F*.

**Figure supplement 1—source data 3.** BNST/CeA/BLP action potential inter-event intervals, as shown in *Figure 6—figure supplement 1G*.

**Figure supplement 1—source data 4.** CeA baseline cell properties by cluster, as shown in *Figure 6—figure supplement 1H*.

**Figure supplement 1—source data 5.** BLP baseline cell properties by cluster, as shown in *Figure 6—figure supplement 1I*.

**Figure supplement 1—source data 6.** Effect of DRN$^{DAT}$ input on CeA cell properties by cluster (EPSPs/IPSPs), as shown in *Figure 6—figure supplement 1J*.

**Figure supplement 1—source data 7.** Effect of DRN$^{DAT}$ input on CeA total voltage area by cluster, as shown in *Figure 6—figure supplement 1J*.

**Figure supplement 1—source data 8.** Effect of DRN$^{DAT}$ input on BLP cell properties by cluster (EPSPs/IPSPs), as shown in *Figure 6—figure supplement 1K*.

**Figure supplement 1—source data 9.** Effect of DRN$^{DAT}$ input on BLP total voltage area by cluster, as shown in *Figure 6—figure supplement 1K*.

**Figure supplement 1—source data 10.** CeA/BLP ex vivo baseline cell properties used for hierarchical clustering, as shown in *Figure 6—figure supplement 1L, M*.

(*Figure 7B*). We confirmed ex vivo that blue light delivery alone onto DRN$^{DAT}$ terminals did not elicit a ChrimsonR-mediated postsynaptic potential in CeA neurons (*Figure 7—figure supplement 1B–F*), and that red light delivery was still capable of eliciting ChrimsonR-mediated EPSPs and IPSPs during continuous delivery of blue light (*Figure 7—figure supplement 1G–I*).

We then performed microendoscopic epifluorescent calcium imaging during the three-chamber sociability task where mice freely explored a chamber containing a novel juvenile mouse and a novel object at opposite ends. Given that social isolation produces changes in long-term potentiation of synapses onto DRN$^{DAT}$ neurons (*Matthews et al., 2016*), we were limited to a single manipulation of social isolation for each mouse. We hypothesized that stimulation of DRN$^{DAT}$ inputs to CeA

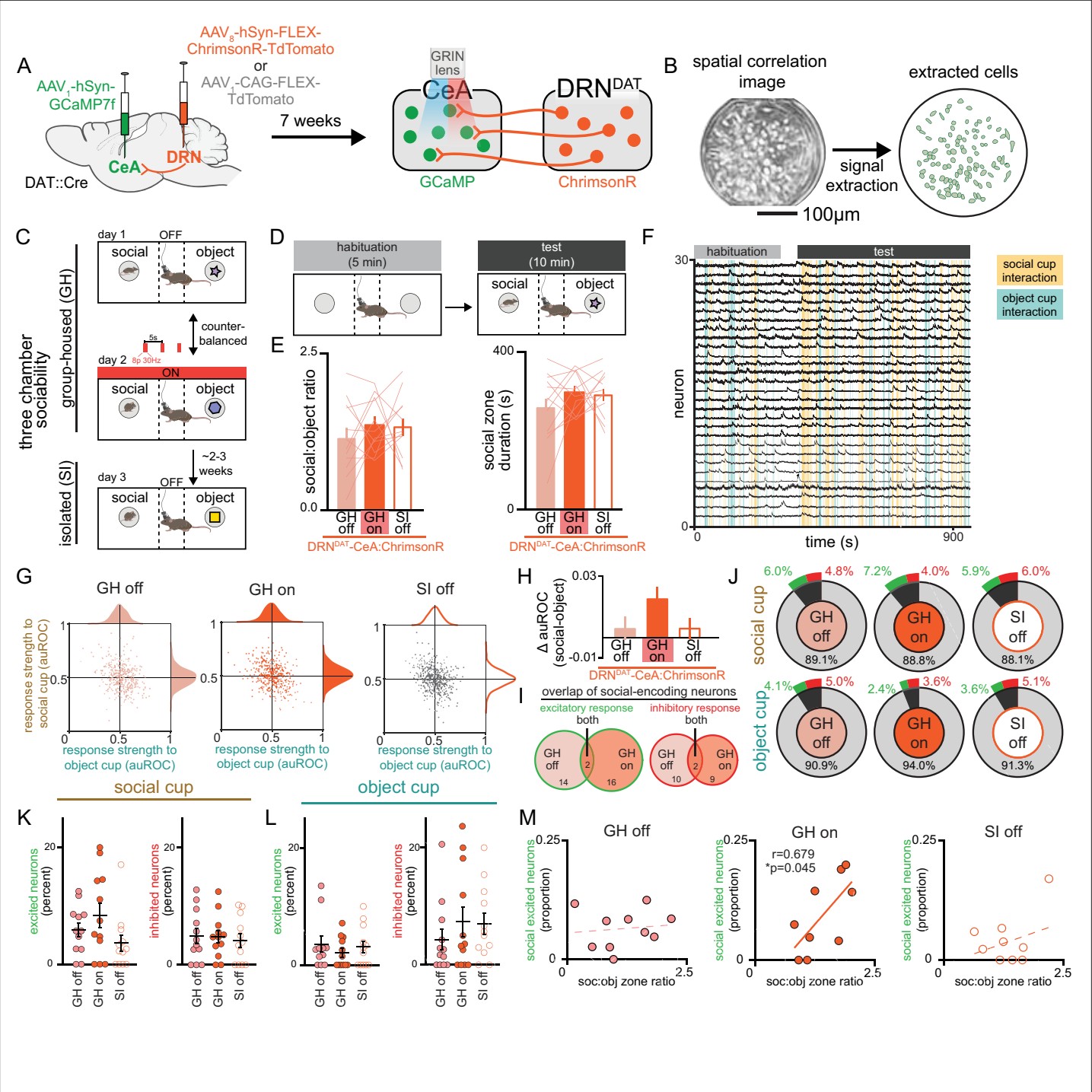

**Figure 7.** Simultaneous calcium imaging of CeA neurons and optogenetic stimulation of DRN^DAT terminals in CeA. (**A**) AAV₁-hSyn-GCaMP7f was injected into the CeA and AAV₈-hSyn-FLEX-ChrimsonR-TdTomato or AAV₁-CAG-FLEX-TdTomato was injected into the DRN of DAT-Cre mice, and a GRIN lens was implanted over CeA. Experiments were conducted 7 weeks following surgery to allow adequate virus expression in axon terminals. (**B**) Example spatial correlation image and extracted ROIs of CeA neurons following calcium imaging processing. (**C**) Three chamber sociability paradigm. While group-housed, mice explored a three-chamber apparatus with a novel male juvenile stimulus on one side and a novel object stimulus on the other. During one day of the imaging experiment, DRN^DAT terminals were not stimulated, and in another session, DRN^DAT terminals were stimulated with red light delivery. Mice underwent a third imaging session, without photostimulation, following 24 hr of social isolation. (**D**) Mice first explored the three-chamber apparatus without social or object stimuli for a 5-min habituation period, then with the social and object stimuli for a 10-min test period. (**E**) Social:object ratio (left) and total social cup interaction time (right) during GH stimulation and no stimulation sessions and 24 hr isolated session in mice expressing ChrimsonR in DRN^DAT neurons. Bar and line graphs represent mean ± SEM (N=12 mice; mixed-effects model: $F_{1.897,30.36}=0.5767$,

*Figure 7 continued on next page*

*Figure 7 continued*

p=0.5591). (**F**) Representative traces from CeA calcium imaging during one three-chamber imaging session. (**G**) Scatter and distribution plots indicating the response strength (auROC) of recorded CeA neurons to social and object cups (GH off: n=429 cells, N=15 mice; GH on: n=441 cells, N=15 mice; SI off: n=484 cells, N=15 mice).(**H**) Difference in response strength (Δ auROC) of CeA neurons to social and object cups (GH off: n=429 cells; GH on: n=441 cells; SI off: n=484 cells) Kruskal-Wallis test: K-W statistic: 6.172, *p=0.0457; Dunn's multiple comparisons test: GH off vs GH on—p=0.0580, GH off vs SI off—p>0.9999. (**I**) Venn diagrams showing overlap of social-encoding neurons (displaying an excitatory response, left, or an inhibitory response, right as defined with auROC) in GH off and GH on sessions (GH off and GH on co-registered neurons: n=202 cells). 16 co-registered GH off cells and 18 GH on cells exhibited an excitatory response to social stimulus with 2 cells having the same response across conditions, whereas 12 co-registered GH off and 11 GH on cells exhibited an inhibitory response with 2 cells having the same response across conditions. (**J**) Proportion of CeA neurons responsive to social and object cups, further classified as an excitatory (green) or inhibitory (red) response to the stimulus as defined with auROC. (**K**) Proportion of recorded neurons that have an excitatory or inhibitory response to the social cup and (**L**) to the object cup (N=12 mice). (**M**) Correlation between social preference in three-chamber task and the proportion of CeA neurons that have an excitatory response to the social cup. The proportion of socially excited neurons is positively correlated with soc:obj zone ratio only for the GH on condition (Pearson correlation: *r*=0.6785, p=0.0445, N=9 mice). Bar and line graphs show mean ± SEM. *p<0.05.

The online version of this article includes the following source data and figure supplement(s) for figure 7:

**Source data 1.** DRN[DAT]-CeA:ChrimsonR three-chamber social:object ratio and social zone duration, as shown in *Figure 7E*.

**Source data 2.** CeA response strength to social and object stimuli, as shown in *Figure 7G*.

**Source data 3.** CeA response strength (change in auROC, social – object), as shown in *Figure 7H*.

**Source data 4.** CeA response overlap of social-encoding neurons, as shown in *Figure 7I*.

**Source data 5.** CeA response classification to social and object stimuli, as shown in *Figure 7J*.

**Source data 6.** Percentage of CeA neurons excited/inhibited by social stimulus, as shown in *Figure 7K*.

**Source data 7.** Percentage of CeA neurons excited/inhibited by object stimulus, as shown in *Figure 7L*.

**Source data 8.** Proportion of CeA neurons excited by social stimulus x social:object ratio, as shown in *Figure 7M*.

**Figure supplement 1.** Ex vivo validation of simultaneous calcium imaging and photostimulation and behavioral and neural effects of DRN[DAT]-CeA:TdTomato stimulation in CeA.

**Figure supplement 1—source data 1.** CeA ex vivo EPSP/IPSP voltage peak in response to 635 nm or 470 nm wavelength light, as shown in *Figure 7—figure supplement 1E*.

**Figure supplement 1—source data 2.** CeA ex vivo EPSP/IPSP voltage area in response to 635 nm or 470 nm wavelength light, as shown in *Figure 7—figure supplement 1F*.

**Figure supplement 1—source data 3.** CeA ex vivo EPSP/IPSP voltage peak in response to just 635 nm or simultaneous 635 nm and 470 nm wavelength light, as shown in *Figure 7—figure supplement 1H*.

**Figure supplement 1—source data 4.** CeA ex vivo EPSP/IPSP voltage area in response to just 635 nm or simultaneous 635 nm and 470 nm wavelength light, as shown in *Figure 7—figure supplement 1I*.

**Figure supplement 1—source data 5.** DRN[DAT]-CeA:TdTomato three-chamber social:object ratio and social zone duration, as shown in *Figure 7—figure supplement 1J, K*.

**Figure supplement 1—source data 6.** CeA response strength (change in auROC, social – object), as shown in *Figure 7—figure supplement 1L*.

**Figure supplement 1—source data 7.** Proportion of CeA neurons excited by object stimulus x social:object ratio, as shown in *Figure 7—figure supplement 1N*.

**Figure supplement 1—source data 8.** Proportion of CeA neurons inhibited by social stimulus x social:object ratio, as shown in *Figure 7—figure supplement 1O*.

**Figure supplement 2.** Ex vivo validation of simultaneous calcium imaging and photostimulation and behavioral and neural effects of DRN[DAT]-CeA:TdTomato stimulation in CeA.

would mimic a loneliness-like state, consistent with our ChR2 manipulations with group-housed mice. Thus, we compared three conditions —group-housed without DRN[DAT] stimulation (GH off), group-housed with DRN[DAT] stimulation (GH on), and 24 hr socially isolated without DRN[DAT] stimulation (SI off; *Figure 7C, D*) to allow for within-subjects comparisons.

In contrast to the photostimulation experiments in *Figure 4*, here we aimed to investigate the impact of DRN[DAT] neuron stimulation on neural dynamics within the CeA without inducing robust behavioral changes that could introduce sensorimotor confounds to changes in neural activity due to stimulation. We successfully optimized viral expression and illumination parameters to minimize changes in social preference with DRN[DAT]-CeA with ChrimsonR to prioritize comparison of the neural

dynamics (*Figure 7E*) and also did not observe any behavioral effects of illumination in TdTomato-expressing mice (*Figure 7—figure supplement 1J, K*).

We then aligned the recorded CeA calcium traces with social cup and object cup interactions and found a striking diversity of neuronal responses to these stimuli under the three experimental conditions (*Figure 7—figure supplement 2A, B*). We next determined the response strength of individual CeA neurons to either stimulus under the three conditions (*Figure 7F, G*) using an area under ROC curve-based approach (*Kingsbury et al., 2019*; *Li et al., 2017*) to determine responsiveness of CeA neurons to social and object stimuli (*Figure 7—figure supplement 1M*). At a single-cell level, we did not observe significant changes in CeA response strength or proportion of neurons significantly responding to social or object stimuli across the three conditions (*Figure 7G, H*). However, we did find a trend indicating stronger responses toward social stimuli compared to object stimuli in the GH on condition compared to the GH off condition in mice expressing ChrimsonR (*Figure 7J*), but not TdTomato (*Figure 7—figure supplement 1L*) in DRN^DAT neurons. Importantly, in co-registered neurons, we found little overlap between CeA neurons excited by the social stimulus in both GH on and GH off conditions (*Figure 7I*), suggesting that DRN^DAT terminal stimulation may recruit separate ensembles of CeA neurons to represent social stimuli. Considering the variability in social preference behavior across mice and the diverse effects of photostimulation depending on the mouse's social history, we next considered the responses of CeA neurons to social and object stimuli on an animal-by-animal basis. While we did not observe significant changes in the proportion of excitatory or inhibitory responses to social or object stimuli across the three conditions (*Figure 7K, L*), we did find a significant positive correlation between the proportion of socially excited CeA neurons and social preference in the GH on condition, but not in the GH off or SI off conditions (*Figure 7M*). Importantly, we do not observe a correlation between social preference and object-excited CeA neurons (*Figure 7—figure supplement 1N*) or socially inhibited CeA neurons (*Figure 7—figure supplement 1O*). This result may suggest that DRN^DAT input in the CeA in a behaviorally relevant task allows for a functional shift in its dynamics that enables it to predict the amount of social preference the mouse exhibits.

## DRN^DAT-CeA photoinhibition blocks isolation-induced sociability

Finally, considering that the DRN^DAT-CeA projection is sufficient in promoting sociability, we next assessed whether activity in the DRN^DAT-CeA projection is necessary for the rebound in sociability that occurs following acute social isolation (*Matthews et al., 2016*). We injected an AAV enabling Cre-dependent expression of NpHR into the DRN of DAT::Cre male mice and implanted optic fibers over the BNST, CeA, or BLP (*Figure 8A*). We allowed 7 weeks for adequate terminal expression, after which we inhibited DRN^DAT terminals in the BNST, CeA, and BLP while mice performed the three-chamber sociability task (*Figure 8B*). Inhibition of DRN^DAT terminals in downstream regions while mice were group-housed did not change social preference (*Figure 8—figure supplement 1A, B*). However, inhibition of DRN^DAT terminals in CeA, but not BNST or BLP, blocked the rebound in sociability associated with acute social isolation (*Figure 8C*). Additionally, we found that optically inhibited changes in social preference in DRN^DAT-CeA mice were negatively correlated with social dominance (*Figure 8D*), suggesting that the DRN^DAT-CeA projection is necessary for the expression of isolation-induced social rebound in a rank-dependent manner.

## Discussion

Neural circuits that motivate social approach are essential in maintaining social connections and preventing isolation. Here we show that DRN^DAT neurons can exert a multi-faceted influence over behavior, with the pro-social effects mediated by the projection to the CeA, the avoidance effects mediated by the projection to the BLP, and the pro-exploratory effects mediated by the projection to the BNST. Our data suggest these effects are enabled via separable functional projections, dense collateralization, co-transmission, and precisely organized synaptic connectivity. Notably, our experiments were conducted in male mice; future work should investigate whether similar circuit mechanisms operate in females and explore the biological basis of sex-specific responses to social isolation (*Yang et al., 2013*; *Zilkha et al., 2021*). In addition, while terminal photostimulation allowed projection-specific manipulation, it may have inadvertently activated fibers of passage—a limitation that could

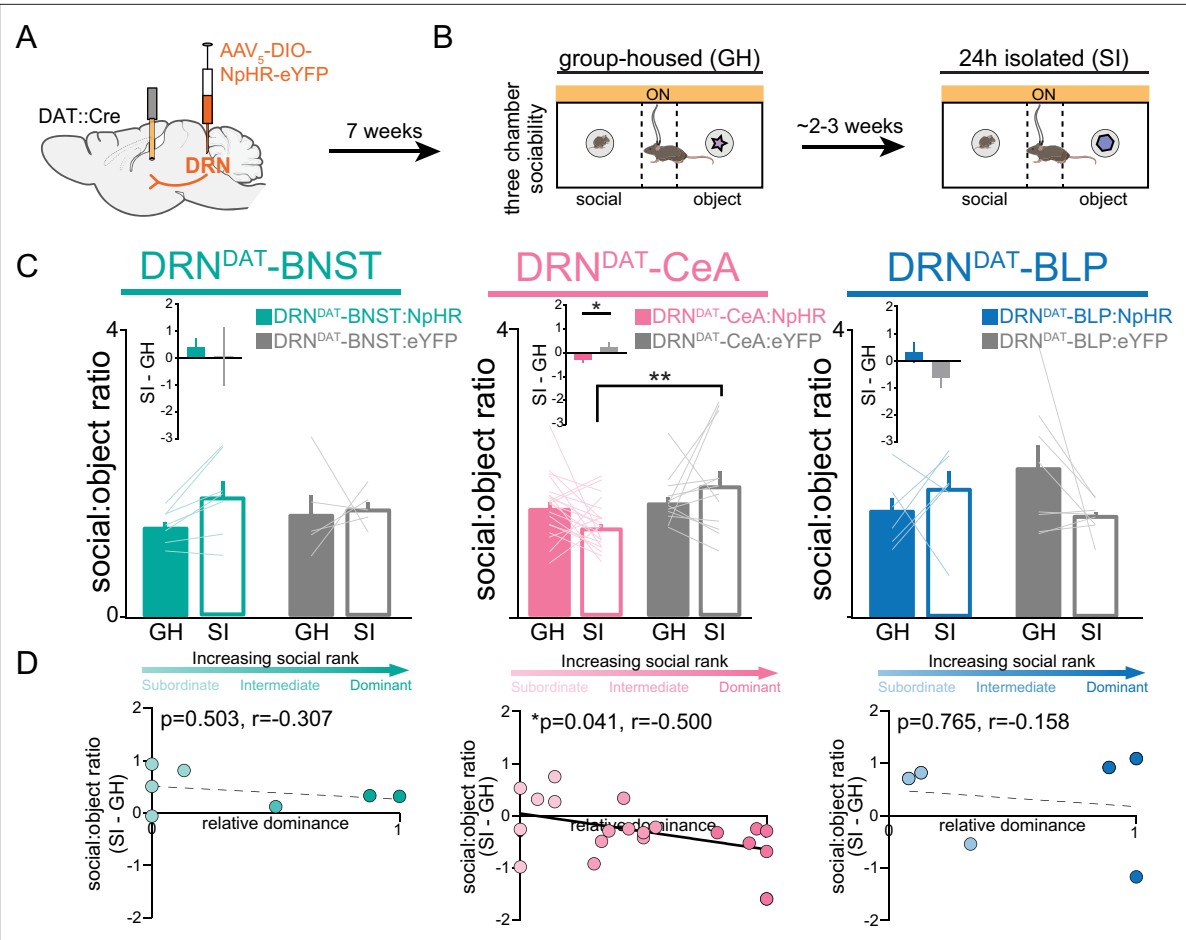

**Figure 8.** DRN$^{DAT}$-CeA photoinhibition blocks isolation-induced sociability. (**A**) AAV$_5$-DIO-NpHR-eYFP or AAV$_5$-DIO-eYFP was injected into the DRN of DAT::Cre mice and optic fibers implanted over the BNST, CeA, or BLP to photoinhibit DRN$^{DAT}$ terminals. (**B**) After >7 weeks for viral expression, mice were assayed on the three-chamber sociability assay with delivery of continuous yellow light for photoinhibition, once when group-housed and once following 24 hr of social isolation (2–3 weeks after the initial session). (**C**) Photoinhibition of DRN$^{DAT}$-BNST terminals in NpHR-expressing mice (DRN$^{DAT}$-BNST:NpHR) had no significant effect on time spent in the social zone relative to the object zone (DRN$^{DAT}$-BNST:NpHR: N=7 mice, DRN$^{DAT}$-BNST:eYFP: N=5 mice; 'social:object ratio'; two-way RM ANOVA: light x group interaction, $F_{1,10}$=1.005, p=0.3397), but reduced social:object ratio for isolated DRN$^{DAT}$-CeA:NpHR mice compared to isolated DRN$^{DAT}$-CeA:eYFP mice (DRN$^{DAT}$-CeA:NpHR: N=20 mice, DRN$^{DAT}$-CeA:eYFP: N=12 mice; 'social:object ratio'; two-way RM ANOVA: light x group interaction, $F_{1,30}$=4.909, p=0.0344; multiple comparisons test, DRN$^{DAT}$-CeA:NpHR$^{SI}$ vs DRN$^{DAT}$-CeA:eYFP$^{SI}$ adjusted **p=0.0017). In addition, terminal photoinhibition had no effect for DRN$^{DAT}$-BLP:NpHR mice (DRN$^{DAT}$-BLP:NpHR: N=6 mice, DRN$^{DAT}$-BLP:eYFP: N=8 mice; 'social:object ratio'; two-way RM ANOVA: light x group interaction, $F_{1,12}$=3.346, p=0.0923). Inset bar graphs show the difference in social:object ratio in isolated and grouped conditions. A significant difference between NpHR$^{CeA}$ and eYFP$^{CeA}$ groups was observed (unpaired t-test: $t_{29}$=2.177, p=0.0377). (**D**) Scatter plots displaying relative dominance plotted against the change in social zone time (isolated-grouped), showing significant negative correlation in NpHR$^{CeA}$ mice (Pearson's correlation: r=−0.500, p=0.0414, N=20 mice). Bar and line graphs show mean ± SEM. *p<0.05, **p<0.01.

The online version of this article includes the following source data and figure supplement(s) for figure 8:

**Source data 1.** DRN$^{DAT}$-ALL:NpHR three-chamber social:object ratio (GH on and SI on), as shown in *Figure 8C*.

**Source data 2.** DRN$^{DAT}$-ALL:NpHR three-chamber social:object ratio (SI – GH) x relative dominance, as shown in *Figure 8D*.

**Figure supplement 1.** Photoinhibition of DRN$^{DAT}$-BNST:NpHR, DRN$^{DAT}$-CeA:NPHR, DRN$^{DAT}$-BLP:NpHR terminals does not affect social preference in group-housed mice.

**Figure supplement 1—source data 1.** DRN$^{DAT}$-ALL:NpHR three-chamber social:object ratio (GH off and GH on), as shown in *Figure 8—figure supplement 1B*.

be addressed in future studies using intersectional genetic strategies. Despite these caveats, these observed circuit features may facilitate a coordinated, but flexible, response in the presence of social stimuli that can be flexibly guided based on internal social homeostatic need state.

## DRN^DAT circuit arrangement enables a broadly distributed, coordinated response

Our findings revealed several features of the DRN^DAT circuit which might facilitate a concerted response to novel social and non-social situations. Firstly, we observed dissociable roles for discrete downstream projections – a common motif of valence-encoding neural circuits (*Tye, 2018*). Biased recruitment of these 'divergent paths' *Tye, 2018* to the BNST, CeA, and BLP by upstream inputs may serve to fine-tune the balance between social investigation and environmental exploration: facilitating behavioral flexibility with changing environmental conditions or internal state. Secondly, we demonstrate extensive collateralization of DRN^DAT neurons. In other populations, collateralization is proposed to aid temporal coordination of a multifaceted response: enabling synchronous activation of distributed regions (*Rockland, 2018*; *Waselus et al., 2011*). This feature may, therefore, facilitate coordinated recruitment of the BNST and CeA, allowing these regions to work in concert to promote social approach while also maintaining vigilance to salient environmental stimuli. Thirdly, we find precise synaptic organization in the DRN^DAT modulation of downstream neuronal activity that allows for qualitatively distinct response profiles of downstream targets. Combined with the spatially segregated downstream receptor expression pattern, this organization may allow DRN^DAT neurons to elicit broad, yet finely tuned, control over the pattern of neuronal activity, on multiple timescales—perhaps explaining the diverse behavior effects between the BNST- and CeA- projecting DRN^DAT populations, despite heavy collateralization.

Although we hypothesized that stimulating DRN^DAT inputs to the CeA in group-housed mice would mimic a state similar to that of isolation, we did not observe that the isolation OFF condition produced neural responses more similar to the group-housed ON condition. This suggests two possibilities: (1) that the photostimulation impacted neural activity beyond the endogenous dopamine innervation that may occur with social isolation because it is more potent of a change or (2) that the timing of endogenous dopamine innervation is different and is partially quenched upon exposure to a social agent. To completely understand the temporal dynamics of dopamine signaling with isolation and the firing of DRN^DAT neurons upon isolation, further experiments will require exploration of DRN^DAT stimulation parameter space, endogenous neural activity in the DRN^DAT-CeA circuit during social isolation, and the effects of DRN^DAT stimulation timing.

## Separable projections mediate social behavior and valence

Our data support the hypothesis that separable DRN^DAT projections mediate distinct functional roles: a feature which has been previously observed in other neuronal circuits (e.g. *Han et al., 2017*; *Kim et al., 2013*; *Kohl et al., 2018*; *Lammel et al., 2011*; *Namburi et al., 2015*). The DRN^DAT circuit attributes we describe above may further enable this system to modulate other diverse forms of behavior (e.g. arousal *Cho et al., 2017*, fear/reward associations *Lin et al., 2020*; *Groessl et al., 2018*, and antinociception *Li et al., 2016*; *Meyer et al., 2009*; *Yu et al., 2021*). These could be mediated via other downstream projections and/or via these same projections under different environmental contexts, testing conditions, social histories, and/or internal states. Further work is required to determine how this system is able to exert a broad influence over multiple forms of behavior. Indeed, a recent study examined DRN^DAT projection to the nucleus accumbens and its role in promoting sociability (*Choi et al., 2022*), suggesting a parallel circuit to that described in the current study. Collectively, however, our data and others support a role for the DRN^DAT system in exerting a coordinated behavioral response to novel situations – both social and non-social.

The CeA has been implicated in mediating the response to threats – orchestrating defensive behavioral responses and autonomic changes via efferents to subcortical (*Davis et al., 2010*; *Fadok et al., 2018*; *Gungor and Paré, 2016*) and brainstem nuclei (*Tovote et al., 2016*). One possible interpretation, therefore, is that DRN^DAT input to the CeA suppresses fear-promoting neuronal ensembles in order to facilitate social approach. In the maintenance of social homeostasis, suppression of fear in the presence of social stimuli may represent an adaptive response – preventing salient social stimuli from being interpreted as a threat. Indeed, other need states, such as hunger,

are associated with fear suppression and higher-risk behavior (*Padilla et al., 2016*), suggesting a conserved response to homeostatic imbalance (*Matthews and Tye, 2019*). However, the motivation to attend to social stimuli may also be driven by territorial defense (interacting with social rank), highlighting a need to further understand how internal states can play into the output of this system. A more comprehensive knowledge of the functional cell types modulated by DRN$^{DAT}$ activity will facilitate our understanding of how this input can shape the downstream neuronal representation of social and non-social stimuli.

In contrast to the CeA, photoactivation of the DRN$^{DAT}$-BLP projection produced avoidance of the stimulation zone, suggesting an aversive state. This differs from the valence-independent role of VTA dopamine input to the greater BLA complex, wherein dopamine signaling gates synaptic plasticity for associative learning of both positive and negative valence (*Tye et al., 2010*) and responds to salient stimuli predicting both positive and negative outcomes (*Lutas et al., 2019*). However, DRN and VTA axonal fields differ within the BLA complex, with DRN$^{DAT}$ terminals being more concentrated within the BLP, and VTA$^{DAT}$ inputs traversing the LA, BLA, and intercalated cells more densely.

While there have been seemingly contradictory reports on the effect of dopamine on excitability in the BLA (*Tye et al., 2010*; *Bissière et al., 2003*; *Lutas et al., 2019*), our observations using photostimulation of DRN$^{DAT}$ terminals (in short phasic bursts) are consistent with in vivo extracellular recordings combined with electrical stimulation of the midbrain (*Rosenkranz and Grace, 1999*). One unifying hypothesis is that dopamine induces an *indirect* GABA-mediated suppression of pyramidal neurons, which may attenuate their response to weak inputs, while *directly* exciting pyramidal neurons to augment their response to large inputs (*Kröner et al., 2005*; *Rosenkranz and Grace, 1999*). In this way, amygdala dopamine may underlie a similar role to cortical dopamine (*Vander Weele et al., 2018*; *Gulledge and Jaffe, 2001*) – enhancing signal-to-noise ratio to facilitate behavioral responses to salient stimuli (*Vander Weele et al., 2018*).

A similar complexity surrounds dopamine's effects in the BNST. Photostimulation of DRN$^{DA}$-BNST projections has been shown to have antinociceptive effects on male mice upon formalin injection into the paw, whereas females do not show this antinociceptive effect and instead display contextual hyperlocomotion (*Yu et al., 2021*). Given that our study does not include females, there are likely behavioral and physiological sex differences that beg further investigation. Additionally, prior studies using optogenetic stimulation (*Li et al., 2016*; *Yu et al., 2021*) or exogenously applied dopamine (*Krawczyk et al., 2011*; *Maracle et al., 2019*) report a wide range of outcomes on BNST excitability, including a higher frequency of inhibitory responses that we observed. Notably, one study found that higher doses of exogenously applied dopamine decreased the amplitude of GABA$_A$-IPSC in ovBNST neurons (*Krawczyk et al., 2011*), potentially consistent with our results. Interestingly, animals given intermittent access to sucrose displayed a significant increase in GABA$_A$-IPSCs in ovBNST neurons following dopamine application (*Maracle et al., 2019*), raising the possibility that experience-dependent factors shape dopamine sensitivity in the BNST, paralleling how dopaminergic modulation in the cortex may be state-dependent (*Vander Weele et al., 2018*). Considering that our ex vivo recordings were performed in group-housed mice, future work should test whether social isolation alters ovBNST responses to dopamine, providing a framework for understanding how environmental states influence dopaminergic neuromodulatory tone.

## Multi-transmitter phenotype of DRN$^{DAT}$ neurons may permit modulation on different timescales

DRN$^{DAT}$ neurons possess an impressive repertoire of signaling molecules: alongside dopamine and glutamate subsets of DRN$^{DAT}$ neuron express VIP and NPW (*Dougalis et al., 2012*; *Huang et al., 2019*; *Motoike et al., 2016*). While there is some partial segregation of VIP- and NPW-expressing neurons (*Huang et al., 2019*), our receptor expression analyses suggest that these neuropeptides converge on the same neurons in the BNST and CeA. This co-localization is intriguing, given that *Vipr2* is typically coupled to the excitatory G$_s$-protein (*White et al., 2010*), while *Npbwr1* is coupled to the inhibitory G$_i$-protein (*Nagata-Kuroiwa et al., 2011*; *Tanaka et al., 2003*). Therefore, signaling through these receptors may exert opposing actions on downstream cells. Recruitment of neuropeptidergic signaling pathways may support slower, sustained downstream modulation, for example, in hunger-mediating hypothalamic Agouti-Related Peptide (AgRP) neurons, neuropeptide co-release is essential for sustaining feeding behavior (*Chen et al., 2019*). Therefore, a delayed, persistent

neuropeptide-mediated signal might enable downstream modulation to outlive phasic DRN[DAT] activity: promoting behavioral adjustments over longer timescales.

While the functional role of these neuropeptides remains to be elucidated, studies on knockout mice suggest a role for NPW in social behavior and stress responding (*Nagata-Kuroiwa et al., 2011*; *Motoike et al., 2016*). Furthermore, in humans with a single-nucleotide polymorphism (SNP) of the *NPBWR1* gene (which impairs receptor function), the perception of fearful/angry faces was more positive and less submissive (*Watanabe et al., 2012*), suggesting a possible role for NPW signaling in interpreting social signals. Similarly, the function of DRN VIP +neurons has received little attention in rodent models, but there has been more focus on the role of VIP in avian social behavior (*Kingsbury and Wilson, 2016*). Of particular interest, in the rostral arcopallium (a homolog of mammalian amygdala *Reiner et al., 2004*), VIP binding density is elevated in birds during seasonal flocking (*Wilson et al., 2016*). This suggests that elevated VIP receptor expression may encourage affiliative social grouping behavior in birds (*Wilson et al., 2016*). Thus, NPW and VIP may act in concert with fast glutamate-mediated and slow dopamine-mediated neurotransmission in the central extended amygdala to modulate behavior on different timescales.

Together, these findings suggest that NPW and VIP may act in concert with fast glutamate and slower dopamine signaling to shape social behavior across multiple timescales within the extended amygdala. Although we observed relatively low expression of *Drd1* in the BNST and CeA compared to *Vipr2* and *Npbwr1*, it remains possible that dopamine's effects are mediated by other receptors, such as *Drd3*, *Drd4*, and *Drd5*—warranting further investigation.

## Conclusion

Together, these findings reveal that DRN[DAT] projections exhibit substantial functional specialization, with anatomically distinct pathways modulating different facets of behavior. The DRN[DAT]-CeA projection promotes sociability, DRN[DAT]-BLP input drives avoidance, and DRN[DAT]-BNST enhances vigilant exploration, highlighting the diverse roles of this neural circuit in coordinating adaptive responses to social and environmental contexts. These findings uncover a circuit mechanism through which DRN[DAT] projections orchestrate distinct behavioral features of a loneliness-like state, providing a framework for understanding how neuromodulatory systems guide complex social and emotional behaviors and suggesting potential targets for therapeutic intervention in affective disorders.

# Materials and methods

## Key resources table

| Reagent type (species) or resource | Designation | Source or reference | Identifiers | Additional information |
|---|---|---|---|---|
| Strain, strain background (*Mus musculus, male*) | DAT[IREScre] (B6.SJL-Slc6a3[tm1.1(cre)Bkmn]/J) | Jackson Laboratory | IMSR_JAX: 000664 | |
| Strain, strain background (*M. musculus, male*) | C57BL/6 J | Charles River Laboratories | Strain code 631 | |
| Strain, strain background (*AAV*) | AAV5-EF1α-DIO-ChR2-eYFP | University of North Carolina Vector Core | | https://www.med.unc.edu/genetherapy/vectorcore/in-stock-aav-vectors/deisseroth/ |
| Strain, strain background (*AAV*) | AAV5-EF1α-DIO-eYFP | University of North Carolina Vector Core | | https://www.med.unc.edu/genetherapy/vectorcore/in-stock-aav-vectors/deisseroth/ |
| Strain, strain background (*AAV*) | AAV9-Syn-ChrimsonR-tdTomato | University of North Carolina Vector Core | | https://www.med.unc.edu/genetherapy/vectorcore/in-stock-aav-vectors/deisseroth/ |
| Strain, strain background (*AAV*) | HSV-LS1L-mCherry-IRES-flpo | Viral Gene Transfer Core Facility at MIT | | https://biology.mit.edu/faculty-and-research/core-facilities/ |

*Continued on next page*

*Continued*

| Reagent type (species) or resource | Designation | Source or reference | Identifiers | Additional information |
|---|---|---|---|---|
| Strain, strain background (*AAV*) | AAV5-EF1α-DIO-eNpHR3.0-eYFP | Addgene | RRID:Addgene_26966 | |
| Strain, strain background (*AAV*) | AAV1-syn-jGCaMP7f | Addgene | RRID:Addgene_104488 | |
| Strain, strain background (*AAV*) | AAV1-CAG-TdTomato | UPenn vector core | | https://www.med.upenn.edu/carot/aav-core.html |
| Chemical compound | CTB-555 | ThermoFisher | C34776 | |
| Chemical compound | CTB-647 | ThermoFisher | C34778 | |
| Chemical compound | DAPI | Sigma-Aldrich | D9542 | 1:50,000 |
| Chemical compound | PVA-DABCO | Sigma-Aldrich | 10981 | |
| Chemical compound | Normal Donkey Serum | Jackson Immunoresearch | RRID:AB_2337258 | |
| Chemical compound | CF405-conjugated streptavidin | Biotium | 29032 | 1:1000 |
| Chemical compound, drug | CF633-conjugated streptavidin | Biotium | 29037 | 1:1000 |
| Chemical compound | RNAScope Multiplex Fluorescent Reagent Kit V2 | ACDBio | 323110 | |
| Chemical compound | RNAScope Multiplex Detection Reagents | ACDBio | 323110 | |
| Chemical compound | RNAScope *Drd1a* target probe | ACDBio | 406491-C1 | |
| Chemical compound | RNAScope *Drd2* target probe | ACDBio | 406501-C3 | |
| Chemical compound | RNAScope *Npbwr1* target probe | ACDBio | 547181-C1 | |
| Chemical compound | RNAScope *Vipr2* target probe | ACDBio | 465391-C1 | |
| Chemical compound | TSA fluorophore (green) | PerkinElmer | Fluorescein | |
| Chemical compound | TSA fluorophore (red) | PerkinElmer | Cyanine 3 | |
| Chemical compound | TSA fluorophore (far red) | PerkinElmer | Cyanine 5 | |
| Antibody | Anti-Tyrosine Hydroxylase Antibody | Millipore | RRID:AB_570923 | 1:1000 |
| Antibody | Alexa Fluor 488 AffiniPure Donkey Anti-Chicken IgY (IgG) (H+L) | Jackson ImmunoResearch | RRID:AB_2340375 | 1:1000 |
| Antibody | Alexa Fluor 647-AffiniPure Donkey Anti-Chicken IgY (IgG) (H+L) | Jackson ImmunoResearch | RRID:AB_2340379 | 1:1000 |
| Software | Fluoview software version 4.0 | Olympus | RRID:SCR_014215 | |
| Software | FIJI | ImageJ | RRID:SCR_003070 | |
| Software | CellProfiler | CellProfiler | RRID:SCR_007358 | |
| Software | Adobe Photoshop CC | Adobe | RRID:SCR_014199 | |
| Software | MATLAB | Mathworks | RRID:SCR_001622 | |
| Software | Adobe Illustrator | Adobe | RRID:SCR_010279 | |
| Software | Ethovision XT | Noldus | RRID:SCR_000441 | |
| Software | pClamp 10.4 software | Molecular Devices | RRID:SCR_011323 | |
| Software | GraphPad Prism 8 | Graphpad | RRID:SCR_002798 | |
| Other | Optic fiber | Thor Labs | TS1843490 | |
| Other | Ferrules | Kientec Systems | FSS-LC-330 | |
| Other | Small animal stereotaxic frame | David Kopf Instruments | Model 942 | |
| Other | 0.10 mL Microsyringe | World Precision Instruments (NANOFIL-NF33BL-2) | RRID:SCR_008593 | |
| Other | Microsyringe Pump UMP3 and Controller Micro4 | World Precision Instruments (UMP3-3) | RRID:SCR_008593 | |
| Other | Peristaltic pump for ex vivo recordings | Minipuls 3 Gilson | F155001 | |

*Continued on next page*

*Continued*

| Reagent type (species) or resource | Designation | Source or reference | Identifiers | Additional information |
|---|---|---|---|---|
| Other | Pulse generator | A.M.P.I. Master-8 | RRID:SCR_018889 | |
| Other | Cryostat | Leica biosystems CM3050 S | RRID:SCR_020214 | |
| Other | HM430 Microtome | Thermo Fisher Scientific | RRID:SCR_020020 | |
| Other | Vibrating blade vibratome | Leica Biosystems VT1200 | RRID:SCR_018453 | |
| Other | Confocal Laser-Scanning microscope | Olympus FV1000 | RRID:SCR_020337 | |
| Other | Diode Blue 473 nm Laser | OptoEngine LLC | MBL-III-473/1–100 mW | |
| Other | Horizontal puller for glass microelectrodes for ex vivo recordings | Sutter P-1000 | RRID:SCR_021042 | |
| Other | Multiclamp amplifier for ex vivo recordings | Molecular Devices 700B | RRID:SCR_018455 | |
| Other | Microscope for ex vivo recordings | Scientifica | RRID:SCR_021035 | |
| Other | Microscope for in vivo calcium imaging recordings | Inscopix nVoke System | RRID:SCR_023028 | |

## Animals and housing

All procedures involving animals were conducted in accordance with NIH guidelines and approved by the MIT Committee on Animal Care or the Salk Institute Institutional Animal Care and Use Committee (IACUC protocol #18–00060). DAT$^{IREScre}$ (B6.SJL-Slc6a3$^{tm1.1(cre)Bkmn}$/J) (*Bäckman et al., 2006*) mice were purchased from the Jackson Laboratory (stock no. 006660; the Jackson Laboratory, ME, USA) and bred in-house to generate heterozygous male offspring for experiments. Wild-type C57BL/6 J male mice were purchased from Charles River Laboratories (MA, USA). Mice were housed on a 12 hr:12 hr reverse light-dark cycle (MIT: lights off 9am-9pm; Salk Institute: lights off 9:30am-9:30pm) with food and water available ad libitum. Mice were housed in groups of 2–4 with same-sex siblings. For photoinhibition and CeA calcium imaging experiments, mice were additionally tested following 24 hr of social isolation. Only mice with acceptable histological placements were included in final datasets.

## Surgery and viral constructs

Mice (>7 weeks of age) were anesthetized with isoflurane (inhalation: 4% for induction, ~2% for maintenance, oxygen flow rate 1 L/min) before being placed in a digital small animal stereotax (David Kopf Instruments, CA, USA). Surgeries were performed under aseptic conditions with body temperature maintained by a heating pad throughout. Injections of recombinant adeno-associated viral (AAV) vectors, herpes simplex virus (HSV), or cholera toxin subunit-B (CTB) were performed using a beveled 33-gauge microinjection needle with a 10 µL microsyringe (Nanofil; WPI, FL, USA). Virus or CTB was delivered at a rate of 0.1 µL/min using a microsyringe pump (UMP3; WPI, FL, USA) connected to a Micro4 controller (WPI, FL, USA). Following injection, the needle was maintained in place for ~2 min, then raised up by 0.05 mm and held for ~10 min (to permit diffusion from the injection site) before being slowly withdrawn. Skull measurements were made relative to Bregma for all injections and implants. Implants were secured to the skull by a layer of adhesive cement (C&B Metabond; Parkell Inc, NY, USA) followed by a layer of black cranioplastic cement (Ortho-Jet; Lang, IL, USA). Mice were given preemptive analgesia (1 mg/kg buprenorphine slow-release; subcutaneous; delivered concurrent with warmed Ringer's solution to prevent dehydration), supplemented with meloxicam (1.5 mg/kg; subcutaneous) where necessary, and were monitored on a heating pad until recovery from anesthesia.

AAV$_5$-EF1α-DIO-ChR2-eYFP, AAV$_5$-EF1α-DIO-eYFP, AAV$_5$-EF1α-fDIO-eYFP, and AAV$_9$-Syn-ChrimsonR-tdTomato were packaged by the University of North Carolina Vector Core (NC, USA) and received the AAV$_5$-EF1a-fDIO-eYFP construct from Karl Deisseroth and Charu Ramakrishnan. HSV-LS1L-mCherry-IRES-flpo was packaged by Dr. Rachael Neve at the Viral Gene Transfer Core Facility at MIT (now located at Massachusetts General Hospital). AAV$_5$-EF1α-DIO-eNpHR3.0-eYFP and AAV$_1$-syn-jGCaMP7f were packaged by Addgene (MA, USA), and AAV$_1$-CAG-TdTomato was packaged by the UPenn vector core (PA, USA).

## Immunohistochemistry and confocal microscopy

Mice were deeply anesthetized with sodium pentobarbital (200 mg/kg, intraperitoneal; IP) or euthasol (150 mg/kg; IP) followed by transcardial perfusion with 10 mL ice-cold Ringer's solution and 15 mL

ice-cold 4% paraformaldehyde (PFA). The brain was carefully dissected from the cranial cavity and immersed in 4% PFA for ~6–18 hr before transfer to 30% sucrose solution in phosphate-buffered saline (PBS) at 4 °C. After at least 48 hr, brains were sectioned at 40 µm on a freezing sliding microtome (HM430; Thermo Fisher Scientific, MA, USA) and sections stored at 4 °C in 1 X PBS. For immunohistochemistry, sections were blocked in PBS containing 0.3% Triton X-100 (PBS-T; Sigma-Aldrich, MO, USA) with 3% normal donkey serum (NDS; Jackson Immunoresearch, PA, USA) for 30–60 min at room temperature. This was followed by incubation in primary antibody solution chicken anti-TH (1:1000; AB9702; EMD Millipore, MA, USA) in 0.3% PBS-T with 3% NDS overnight at 4 °C. Sections were then washed in 1 X PBS four times (10 min each) before incubation in secondary antibody solution containing donkey anti-chicken 488 or 647 (1:1000; Jackson Immunoresearch, PA, USA) and a DNA-specific fluorescent probe (DAPI; 1:50,000; Invitrogen, Thermo Fisher Scientific, MA, USA) in 0.2% PBS-T with 3% NDS for 1.5–2 hr at room temperature. Sections were again washed four times in 1 X PBS (10 min each) before being mounted on glass slides and coverslipped using warmed PVA-DABCO (Sigma-Aldrich, MO, USA).

Images were captured on a laser scanning confocal microscope (Olympus FV1000, Olympus, PA, USA) using Fluoview software version 4.0 (Olympus, PA, USA). Images were collected through a 10 X/0.40 NA objective for injection site and optic fiber placement verification, a 20 X/0.75 objective for terminal fluorescence quantification, and an oil-immersion 40 X/1.30 NA objective for neurobiotin-filled neurons and RNAscope analysis (see individual Methods sections for more detail). FIJI *Schindelin et al., 2012*, CellProfiler 3.1 (Broad Institute, MA, USA) *McQuin et al., 2018*, and Adobe Photoshop CC (Adobe Systems Incorporated, CA, USA) were used for subsequent image processing and analysis.

## Downstream fluorescence quantification

In DAT::Cre mice, AAV$_5$-EF1α-DIO-ChR2-eYFP (300 nL) was injected into the DRN (ML:1.20, AP:–4.10, DV:–2.90; needle at a 20° angle from the midline, bevel facing medial) or VTA (ML:0.85, AP:–2.70, DV:–4.50), and after 8 weeks mice underwent perfusion-fixation. Brains were subsequently sectioned at 40 µm, processed with immunohistochemistry for TH and DAPI, and serial z-stack images (3 µm optical thickness) collected at 20 X on a confocal microscope. (See *'Immunohistochemistry and confocal microscopy'* section above for details). A maximum projection was generated in FIJI and background subtraction based on the 'rolling ball' algorithm (radius = 50 pixels) was applied to correct for uneven illumination. The appropriate brain atlas slice *Paxinos and Franklin, 2004*; *Paxinos and Franklin, 2019* was overlaid onto the fluorescent image using the BigWarp plugin (https://imagej.net/BigWarp) *Bogovic et al., 2016* in FIJI, by designating major anatomical landmarks based on DAPI staining and TH expression. Regions of interest (ROIs) were then annotated from the overlaid atlas, and mean fluorescence within each ROI was quantified using FIJI. The PFC was examined from AP: 2.22–1.34, the striatum from AP 1.70–0.74, the BNST from AP 0.37 to –0.11, the CeA from AP –0.82 to –1.94, and the amygdala from AP –0.82 to –2.92. Average images in *Figures 1D, 5A, E and I* were created by aligning individual images (from the middle AP region of the BNST, CeA, or BLP), using the line ROI registration plugin (https://imagej.net/Align_Image_by_line_ROI) in FIJI. An average projection was then performed across all images and the 'royal' LUT applied to visualize relative fluorescence intensity.

## Retrograde tracing and intersectional viral expression

C57BL/6 mice were injected with 150–250 nL CTB conjugated to Alexa Fluor-555 (CTB-555) or Alexa Fluor-647 (CTB-647; Molecular Probes, OR, USA *Conte et al., 2009*) in two of three locations: the BNST (ML:1.10, AP:0.50, DV: –4.30; needle bevel facing back), CeA (ML:2.85, AP: –1.20; DV:–4.75; needle bevel facing back), or BLP (ML:3.35, AP:–2.20, DV:–5.25; needle bevel facing back). To assess retrograde CTB co-expression following injection of both fluorophore-conjugates of CTB into the same region (*Figure 1—figure supplement 2A–C*), injections were either performed sequentially, or CTB-555 and CTB-647 were mixed prior to a single injection. After 7 days to allow for retrograde transport, mice were deeply anesthetized with sodium pentobarbital (200 mg/kg) and perfused-fixed for subsequent histology. Brain sections containing injection sites and the DRN were prepared at 40 µm and processed with immunohistochemistry for TH and DAPI. (See *'Immunohistochemistry and confocal microscopy'* section above for details). CTB injection sites were verified with images acquired on a confocal microscope through a 10 X objective (serial z-stack with 5 µm optical thickness) and

images of the DRN were acquired through a 40 X objective (serial z-stack with 3 μm optical thickness). DRN cells co-expressing CTB and TH were counted manually using the ROI 'point' tool in Fluoview software version 4.0 (Olympus, PA, USA). Counted files were imported into FIJI, and images overlaid onto the appropriate brain atlas image of the DRN using the BigWarp plugin (https://imagej.net/BigWarp; *Bogovic et al., 2016*). The x-y coordinates of counted/marked CTB+/TH + cells were extracted using the 'Measure' function in FIJI. These coordinates were then used to generate heat-maps of cell location (*Figure 1—figure supplement 2D, E*) by creating a 2D histogram using the Matplotlib package *Hunter, 2007* in Python.

Intersectional labeling of the dopaminergic projection from the DRN to the CeA was achieved by injecting HSV-LS1L-mCherry-IRES-flpo (300 nL) into the CeA (ML:2.85, AP:–1.45, DV:–4.55; needle bevel facing medial) and AAV$_5$-fDIO-eYFP (300 nL) into the DRN (ML:1.20, AP:–4.10, DV:–2.90; needle at a 20° angle from the midline, bevel facing medial) of a DAT::Cre mouse. After 8 weeks, mice were perfused-fixed with 4% PFA, and the brain sectioned on a freezing microtome at 40 μm before immunohistochemical processing with TH and DAPI. Images of eYFP-expressing cells in the DRN and terminals in the CeA and BNST were captured on a confocal microscope through a 20 X objective with a serial z-section thickness of 3 μm.

## Behavioral assays and optogenetic manipulations

DAT::Cre mice were injected with 300 nL AAV$_5$- EF1α-DIO-ChR2-eYFP or AAV$_5$-EF1α-DIO-eYFP in the DRN (ML:1.20; AP:–4.10; DV:–2.90; needle at a 20° angle from the right side, bevel facing medial) and optic fibers (300 μm core, NA = 0.37; Thorlabs, NJ, USA), held within a stainless steel ferrule (Precision Fiber Products, CA, USA), were implanted unilaterally or bilaterally over the BNST (unilateral: ML:1.10, AP:0.40, DV:–3.50; bilateral: ML:1.65, AP:0.40, DV:–3.35; 10° angle from midline), CeA (ML:2.85, AP:–1.35, DV:–4.00), or BLP (ML:3.30, AP:–2.20, DV:–4.30). Behavioral experiments commenced 7–8 weeks following surgery for each cohort of mice (18 in total). Mice were handled and habituated to patch cable connection once per day for at least 3 days before beginning optical manipulations. Behavioral testing was performed in a dimly-lit soundproofed room during the mice's active dark phase (~10am-5pm). On each testing day, mice were given at least 1 hr to acclimate to the testing room before experiments began. For optical manipulations, optic fiber implants were connected to a patch cable via a ceramic sleeve (Precision Fiber Products, CA, USA), which itself was connected to a commutator (rotary joint; Doric, Québec, Canada) using an FC/PC adapter, to permit uninhibited movement. The commutator, in turn, was connected via a second patch cable (with FC/PC connectors) to a 473 nm diode-pumped solid-state (DPSS) laser (OEM Laser Systems, UT, USA). To control the output of the laser, a Master-8 pulse stimulator (AMPI, Israel) was used, and the light power set to 10 mW.

### Tube test

Cages of mice (same-sex groups of 2–4) were assayed for social dominance using the tube test *Lindzey et al., 1961*; *Wang et al., 2011*. Mice were individually trained to pass through a clear Plexiglas tube (30 cm length, 3.2 cm inner diameter) over 4 days. Each training trial involved releasing the mouse into the tube from one end and ensuring it traveled through and out the other side. Mice that attempted to reverse, or were reluctant to exit at the other end of the tube, were gently encouraged forwards by light pressure from a plastic stick pressing on their hind region. Between trials, mice freely explored the open arena outside the tube (76x60 cm) for ~30–60 s. Mice performed 8 training trials (4 from each end) on days 1 and 2, and 3 trials (alternating ends) on days 3 and 4. On days 5–8 mice competed against cagemates in a round-robin design. For each contest, mice were released simultaneously into opposite ends of the tube so that they met face-to-face in the center of the tube. The mouse that retreated from the confrontation was designated as the 'loser' and his opponent designated the 'winner'. Across testing days, the side from which animals were released and the order in which they were tested against cagemates was counterbalanced. An animal's 'relative dominance' score reflected their proportion of 'wins' across all contests from 3 to 4 days of testing.

### Open field test (OFT)

The open field was composed of a square arena (51x51 cm) made of transparent Plexiglas with 25 cm high walls. Mice freely explored the arena for 15 min, and blue light (8 pulses with 5ms pulse width, at

30 Hz, every 5 s) was delivered during the middle 5 min epoch of the session. Animals were recorded using a video camera positioned above the arena, and Ethovision XT software was used to track mouse location (Noldus, Netherlands). To assess anxiety-related behavior, for analysis, the chamber was divided into a 'center' square region and a 'periphery', with equal area.

## Three-chamber sociability assay

The apparatus consisted of a 57.5l x 22.5 w x 16.5h cm chamber, with transparent Plexiglas walls and opaque gray plastic floors. The chamber was divided into unmarked left and right compartments (each 23x22.5 cm) and a smaller center compartment (11.5x22.5 cm). An upturned wire mesh cup was placed in the left and right compartments. Each mouse first underwent a habituation session (10 min) where they freely explored the chamber. They were then briefly (~1 min) confined to the center compartment by the insertion of clear Plexiglas walls, while a novel object was placed under one of the two upturned cups, and a juvenile C57BL/6 mouse (3.5–5 weeks of age) was placed under the other upturned cup. The mice were then allowed to freely explore the chamber for a further 10 min. The task was repeated on the second day, with the chamber rotated by 90° relative to external spatial cues, and with a different novel object and novel juvenile mouse. The 10-min test epoch was paired with blue light delivery (8 pulses with 5ms pulse width, at 30 Hz, every 5 s) on one of the 2 days, counterbalanced across animals. Mice were excluded if they showed a strong preference (>70% time spent) for one side of the chamber in the habituation phase, or if they spent more than 1 min on top of the upturned cups during any session. For photoinhibition experiments, the protocol was exactly the same as the ChR2 experiment, except the 10-min test epoch was paired with constant yellow light (589 nm) delivery for 1 day in the 'group-housed' and also during the additional 'isolated' condition. Animals were recorded using a video camera positioned above the chamber and movement tracked using Ethovision XT (Noldus, Netherlands). The social:object ratio reflected the time spent in the 'social' side of the chamber (containing a novel juvenile mouse) divided by the time spent in the 'object' side of the chamber (containing a novel object).

## Juvenile intruder assay

Mice were tested individually in their home cage. They freely explored alone for 5 min after which a novel juvenile mouse was placed in the cage for a further 3 min. The task was repeated on the second day with a different novel juvenile mouse. One of the two sessions was paired with blue light delivery (8 pulses with 5ms pulse width, at 30 Hz, every 5 s) which commenced after 2 min and continued until the end of the task (6 min total). The behavior of the mouse during the 3 min with the juvenile was scored manually using ODLog software (Macropod Software, Australia). Video files were scored twice (by two different observers, blinded to the experimental conditions) and the average of their counts was used for analysis. (See also 'First-order Markov analysis' section).

## Elevated plus maze (EPM)

The EPM was made of gray plastic and consisted of two closed arms (30l x 5 w x 30h cm) and two open arms (30l x 5 w cm), radiating at 90° from a central platform (5x5 cm) and raised from the ground by 75 cm. Mice freely explored for 15 min, with blue light (8 pulses with 5ms pulse width, at 30 Hz, every 5 s) delivered during the middle 5 min epoch of the session. A video camera position above the EPM was used to record animals, and movement was tracked using Ethovision XT (Noldus, Netherlands).

## Real-time place preference (RTPP)

Mice were placed in a 52l x 52 w x 26.5h cm transparent Plexiglas chamber, with clear panels separating left and right sides to leave a 11.5 cm gap for mice to pass through. Mice freely explored for 30 min, during which entry into one side of the chamber resulted in delivery of blue light (15 pulses with 5ms pulse-width, at 30 Hz, every 5 s), which continued until mice exited the zone. Entry into the opposite side did not result in blue light delivery. The side paired with blue light delivery was counterbalanced across animals. A video camera positioned above the arena recorded animals, and mouse movement was tracked using Ethovision XT (Noldus, Netherlands).

## Intra-cranial self-stimulation (ICSS)

Mice were food deprived for 16–20 hr prior to each day of ICSS, in order to encourage behavioral responding. Testing was conducted in an operant chamber (Med Associates, VT, USA) within a custom sound-attenuating outer box. The operant chamber contained two illuminated nose-poke ports, each with an infrared beam, and a cue light positioned above each port. White noise was delivered continuously throughout the session, and successful nose pokes (signaled by a beam break) resulted in an auditory tone (1 s duration, 1 or 1.5 kHz) and illumination of the respective cue light. A nose poke at the 'active' port also triggered delivery of blue light (90 pulses with 5ms pulse width, at 30 Hz) while a nose poke at the 'inactive' port did not trigger light delivery. The physical location of the active and inactive nose-poke ports, and the auditory tone frequency associated with each port, was counterbalanced across animals. On day 1 (training), mice completed a 2 hr session in the operant chamber in which both nose-poke ports were baited with a small amount of palatable food, in order to encourage investigation. On day 2 (testing), mice completed an identical 2 hr session, except the nose-poke ports were not baited. Nose-poke activity was recorded with MedPC software (Med Associates, VT, USA) and subject-averaged cumulative distribution plots were generated using MATLAB (Mathworks, MA, USA). Only data from day 2 was used for analysis.

## Analysis of baseline behavior

The baseline behavior of all mice (i.e without stimulation) was evaluated to uncover any relationships between specific types of behavior assessed in different tasks. These analyses used relative dominance from the tube test, the first 5 min of the OFT and EPM, and the OFF trial from the three-chamber sociability and juvenile intruder assays. Correlation matrices were generated in GraphPad Prism 8 (GraphPad Software, CA, USA) to show the Pearson's correlation coefficient for each pair of variables.

Dimensionality reduction was performed on baseline behavior data using principal component analysis (PCA) with the scikit-learn module *Pedregosa et al., 2011* in Python. The eight input measures from behavioral assays were (1) percent time moving in the OFT, (2) time in the center of the OFT, (3) time in the open arms of the EPM, (4) social:object ratio in the three-chamber assay, and (5) time spent face sniffing, (6) anogenital sniffing, (7) rearing, and (8) grooming in the juvenile intruder assay. The data was first normalized to generate a covariance matrix and then the first 5 PCs were extracted. Relative dominance was concatenated with the resulting PC values for each mouse to color-code individual points in the PC1 vs PC2 plot.

## First-order Markov analysis

Behavioral videos from the juvenile intruder assay were manually annotated so that each second of the 180 s session was assigned a code(s) from 15 behavioral categories:

- - Social behaviors: face sniff (reciprocated), face sniff (non-reciprocated), flank sniff, anogenital sniff (reciprocated), anogenital sniff (non-reciprocated), close follow, approach, dominant climb, attack.
- - Nonsocial behaviors: groom, dig, rear, climb, still, ambulate.

We designed a two-state Markov model, in which behaviors were assigned to either the 'social' or 'nonsocial' categories. For each animal, we created a transition probability matrix from each sequence by counting the number of transitions that occurred and dividing by the total number of occurrences of that behavior. To compute the overall transition probability matrix for the eYFP and ChR2 groups, we took the mean of the transition probability across all individuals in that group. Difference scores between the stimulation OFF and ON sessions were calculated by taking the difference across pairs of transition probability matrices corresponding to each individual, then calculating the mean across eYFP or ChR2-expressing mice.

To verify that a first-order Markov model was an appropriate fit for our data, we computed the log likelihood chi-squared statistic *Gottman and Roy, 1990*:

$$G = 2 \sum_j \sum_i O_{ij} \ln \frac{O_{ij}}{E_{ij}},$$

where $O_{ij} \geq 0$ is the observed number of transitions from state i to j, $E_{ij} \geq 0$ is the expected number of transitions from state i to state j assuming a zeroth order Markov (i.e. no time dependence). We found that G was statistically significant for all subjects in both the 15 state and 2 state models, thus rejecting the null hypothesis of randomly transitioning between states.

We also tested whether a non-stationary model was a better fit for the data than a stationary model. To do this, we divided each subject's behavioral sequence into two segments of equal duration and computed transition probability matrices for each segment. We then computed a variation on the likelihood ratio chi-square statistic *Gottman and Roy, 1990*:

$$LRX = 2 \sum_s \sum_j \sum_i f_{ijs} \ln \frac{\bar{p}_{ijs}}{p_{ij}},$$

where *s* represents the segment, $p_{ij}$ is the probability of transition from state *i* to *j* taken over the entire sequence, $\bar{p}_{ijs}$ is the probability of transition from *i* to *j* for each segment, and $f_{ijs}$ is the number of transitions from state *i* to *j* for each segment. Since not all subjects had a significant difference, we determined that a stationary model was the most appropriate model to fit all our data.

## Ex vivo electrophysiology

DAT::Cre mice received an injection of 300 nL AAV$_5$-DIO-ChR2-eYFP or AAV$_9$-FLEX-ChrimsonR-TdTomato in the DRN (ML:1.20, AP:–4.10, DV:–2.90; needle at a 20° angle from the midline, bevel facing medial), and after at least 8 weeks for transgene expression, mice were deeply anesthetized with sodium pentobarbital (200 mg/kg) or euthasol (150 mg/kg; IP). They were then transcardially perfused with ice-cold (~4 °C) modified artificial cerebrospinal fluid (ACSF; composition in mM: NaCl 87, KCl 2.5, NaH2PO4*H20 1.3, MgCl2*6H2O 7, NaHCO3 25, sucrose 75, ascorbate 5, CaCl2*2H2O 0.5, in ddH20; osmolarity 320–330 mOsm, pH 7.30–7.40), saturated with carbogen gas (95% oxygen, 5% carbon dioxide) before the brain was rapidly and carefully extracted from the cranial cavity. Thick coronal (300 μm) slices containing the BNST, CeA, BLP, and DRN were prepared on a vibrating blade vibratome (VT1200; Leica Biosystems, Germany), in ice-cold modified ACSF saturated with carbogen gas. Brain slices were hemisected with a scalpel blade before transfer to a holding chamber containing ACSF (composition in mM: NaCl 126, KCl 2.5, NaH2PO4*H20 1.25, MgCl2*6H2O 1, NaHCO3 26, glucose 10, CaCl2*H2O 2.4; osmolarity 298–302 mOsm, pH 7.30–7.40) saturated with carbogen, in a warm water bath (~30 °C).

Electrophysiological recordings were commenced after the slices had rested for at least 45 min. During recording, the brain slice was maintained in a bath with continuously perfused ACSF, saturated with carbogen, at 31 ± 1°C using a peristaltic pump (Minipuls3; Gilson, WI, USA). Slices were visualized through an upright microscope (Scientifica, UK) equipped with infrared-differential interference contrast (IR-DIC) optics and a Q-imaging Retiga Exi camera (Q Imaging, Canada). In the BNST, CeA, and BLP, recordings were performed in the region containing fluorescent DRN$^{DAT}$ terminals (expressing ChR2-eYFP or Chrimson-TdTomato) with neurons visualized through a 40 X/0.80 NA water immersion objective. Terminal expression was confirmed by brief illumination from a 470 nm LED light source (pE-100; CoolLED, NY, USA) for ChR2-eYFP, or a metal halide lamp (Lumen 200; Prior Scientific Inc, UK), for ChrimsonR-TdTomato, combined with the appropriate filter set. Borosilicate glass capillaries were shaped on a P-97 puller (Sutter Instrument, CA, USA) to produce pipettes for recording that had resistance values of 3.5–5 MOhm when filled with internal solution composition in mM: potassium gluconate 125, NaCl 10, HEPES 20, MgATP 3, and 0.1% neurobiotin, in ddH20 (osmolarity 287 mOsm; pH 7.3). Whole-cell patch-clamp recordings were made using pClamp 10.4 software (Molecular Devices, CA, USA), with analog signals amplified using a Multiclamp 700B amplifier, filtered at 3 kHz, and digitized at 10 kHz using a Digidata 1550 (Molecular Devices, CA, USA). A 5 mV, 250ms hyperpolarizing step was used to monitor cell health throughout the experiment, and recordings were terminated if significant changes (>20%) occurred to series resistance ($R_s$), input resistance ($R_{in}$), or holding current.

Passive cell properties (capacitance, membrane resistance) were estimated from the current response to hyperpolarizing 5 mV, 250ms steps, delivered in voltage-clamp from a holding potential of –70 mV, using custom MATLAB code written by Praneeth Namburi, based on MATLAB implementation of the Q-Method (*Novák and Zahradník, 2006*). To examine the membrane potential response to current injection, cells were recorded in current-clamp mode, and a series of 1 s steps

were delivered, in 20 pA increments, from –120 pA to 260 pA. The voltage sag amplitude (attributable to the hyperpolarization-activated cation current; $I_h$) was measured as the difference between the peak instantaneous and steady-state membrane potential elicited during a –120 pA step (see *Figure 6L*). The ramp ratio was calculated by dividing the average membrane potential between 900–1000ms by the membrane potential between 100–200ms following step onset, using the largest current step that elicited a subthreshold response (i.e. did not evoke action potentials). The firing delay was taken as the time between current step onset and the first elicited action potential, on delivery of the first current step that was elicited a suprathreshold response (i.e. rheobase current). The max instantaneous firing frequency (max freq.$_{inst}$) was taken as the maximum firing frequency attained during the first 100ms of the depolarizing current steps.

To photostimulate ChR2-expressing DRN$^{DAT}$ terminals in the BNST, CeA, and BLP, 470 nm light was delivered through the 40 X/0.8 NA objective from an LED light source (pE-100; CoolLED, NY, USA). Neurons were recorded at their resting membrane potential in current-clamp mode, and 470 nm light (8 pulses at 30 Hz, 5ms pulse width) was delivered every 30 s. In a minority of cells that showed spontaneous activity at the resting potential, negative current was injected to hold the cell at a subthreshold potential (typically ~–60 mV). The peak amplitude of the optically evoked excitatory post-synaptic potential (EPSP) or trough amplitude of the inhibitory post-synaptic potential (IPSP) was measured from the average trace using Clampfit 10.7 (Molecular Devices, CA, USA), using the 5 s prior to stimulation as baseline. Tau for the decay phase of the IPSP was estimated by fitting the IPSP with a single exponential, from the IPSP trough until return to baseline. Total voltage area was calculated from 0 to 5.5 s following the onset of the first light pulse. In cells where optical stimulation evoked only an EPSP, the response was classed as an 'excitation'; only an IPSP was classed as an 'inhibition', and a combined optically evoked EPSP and IPSP was classed as 'mixed'. To assess the effect of photostimulation on firing activity, constant positive current was injected to elicit consistent spontaneous action potentials, and 470 nm light (8 pulses at 30 Hz, 5ms pulse width) was delivered every 30 s. The interevent interval (IEI) between action potentials was calculated for 5 s before and 5 s after the first pulse of blue light using Clampfit 10.7 (Molecular Devices, CA, USA). A decrease in IEI (indicating an increase in firing rate) was classed as an 'excitation' and an increase in IEI (indicating a decrease in firing rate) was classed as an 'inhibition'.

Following the recording, images showing the location of the recording pipette within the slice were captured through a 4 X/0.10 NA objective. Images were subsequently overlaid onto the appropriate brain atlas image (*Paxinos and Franklin, 2004*; *Paxinos and Franklin, 2019*), recorded cell locations were annotated, and then converted into x-y coordinates in FIJI. Python was used to generate a scatter plot of cell location, with points color-coded by the overall membrane potential response to photostimulation (*Figure 6—figure supplement 1A–C*).

Unsupervised agglomerative hierarchical clustering was used to classify cells according to their baseline electrophysiological properties. This approach organizes objects (in this case cells) into clusters, based on their similarity. The electrophysiological properties used as input features for clustering CeA cells were ramp ratio, max firing frequency, firing delay, and voltage sag, which are characteristics that have been previously shown to distinguish between subtypes of CeA neuron (*Hou et al., 2016*). For clustering BLP cells, we replaced ramp ratio with capacitance, as this measure is often used to distinguish between pyramidal neurons and GABAergic interneurons, which are the two main cell types in this region. Data for each cell property was max-min normalized to produce a 4 x *n* matrix of input features (where n=total number of cells). Clustering was performed using the 'linkage' function of SciPy *Virtanen et al., 2020* in Python, using Ward's linkage method (*Ward, 1963*) and Euclidean distance. Briefly, this approach begins with each cell assigned to a single cluster. Cells that are in closest proximity (i.e. have highest similarity) are then linked to form a new cluster. Then the next closest clusters are linked, and so on. This process is repeated until all cells are included in a single cluster. The output of this analysis is plotted as a hierarchical tree (dendrogram), in which each cell is a 'leaf' and the Euclidean distance on the y-axis indicates the linkage between cells (larger distance indicates greater dissimilarity). To annotate the photostimulation response of cells on the dendrogram (*Figure 6N, P*, *Figure 6—figure supplement 1M*), the response was designated as 'excitation' if action potential IEI decreased with optical stimulation and 'inhibition' if action potential IEI increased on stimulation. If firing data was not available, cells were designated as showing an 'excitation' if only an EPSP was evoked on optical stimulation, and 'inhibition' if only an IPSP was evoked. In cells where

a mixed EPSP/IPSP was elicited, the response was designated as an 'excitation' if the overall voltage area (0–5.5 s following light onset) was positive, and an 'inhibition' if the overall voltage area was negative.

At the end of recording, brain slices were fixed in 4% PFA overnight and then washed in 1 X PBS (4x10 min each). Slices were blocked in 0.3% PBS-T (Sigma-Aldrich, MO, USA) with 3% NDS (Jackson Immunoresearch, PA, USA) for 30–60 min at room temperature. They were then incubated in PBS-T 0.3% with 3% NDS and CF405- or CF633-conjugated streptavidin (1:1000; Biotium, CA, USA) for 90 min at room temperature to reveal neurobiotin labeling. Slices were finally washed four times in 1 X PBS (10 min each) before being mounted on glass slides and coverslipped using warmed PVA-DABCO (Sigma-Aldrich, MO, USA).

## Single molecule fluorescent in situ hybridization (smFISH) with RNAscope

C57BL/6 mice were deeply anesthetized with 5% isoflurane, and brains were rapidly extracted and covered with powdered dry ice for ~2 min. Frozen brains were stored in glass vials at –80 °C before sectioning at 20 µm using a cryostat (CM3050 S; Leica Biosystems, Germany) at –16 °C. Coronal sections were thaw-mounted onto a glass slide, by gentle heating from the underside using the tip of a finger to encourage adhesion of the section to the slide. They were then stored at –80 °C until processing.

Fluorescent in situ hybridization (FISH) was performed using the RNAscope Multiplex Fluorescent assay v2 (Advanced Cell Diagnostics, CA, USA). The following products were used: RNAscope Multiplex Fluorescent Reagent Kit V2 (Catalog #323110), Fluorescent Multiplex Detection Reagents (#323110), target probes for *Mus musculus* genes – *Drd1a* (#406491-C1), *Drd2* (#406501-C3), *Npbwr1* (#547181-C1), and *Vipr2* (465391-C2) – and the Tyramide Signal Amplification (TSA) Plus Fluorescence Palette Kit (NEL760001KT; PerkinElmer Inc, MA, USA) with fluorophores diluted to 1:1000-1:5000. The protocol was performed as recommended by the manufacturer, with some modifications to prevent tissue degradation and optimize labeling specificity in our ROI. Fresh frozen slices were fixed in 4% PFA for 1 hr at 4 °C. Slices were dehydrated in an ethanol series (50%, 70%, 100%, and 100% ethanol, 5 min each) and then incubated in hydrogen peroxide for 8 min at room temperature. Protease treatment was omitted in order to prevent tissue degradation. Slides were then incubated with the desired probes (pre-warmed to 40 °C and cooled to room temperature) for 2 hr at 40 °C in a humidified oven. Following washing (2x30 s in 1 X RNAscope wash buffer), signal amplification molecules (Amp 1, 2, and 3) were hybridized to the target probes in sequential steps, with 30 min incubation for Amp 1 and 2 and 15 min incubation for Amp 3 at 40 °C, all in a 40 °C humidified oven followed by washing (2x30 s in wash buffer). For fluorescent labeling of each amplified probe, slides were incubated in channel-specific HRP for 10 min, followed by incubation with TSA fluorophore (PerkinElmer, MA, USA) for 20 min, and then incubation in HRP-blocker for 10 minutes (with 2x30 s washes between each step). Probes for *Drd1a*, *Drd2*, *Npbwr1*, and *Vipr2* were each labeled with green (TSA Plus Fluorescein), red (TSA Plus Cyanine 3), or far red (TSA Plus Cyanine 5) fluorophores in counterbalanced combinations. Slides were then incubated in DAPI (Advanced Cell Diagnostics, CA, USA) for 10 min, washed in 1 X RNAscope wash buffer, dried for 20 min, coverslipped with warmed PVA-DABCO (Sigma-Aldrich, St. Louis, MO), and left to dry overnight before imaging.

Images were captured on a confocal laser scanning microscope (Olympus FV1000; Olympus, PA, USA) using a 40 X/1.30NA oil immersion objective. Serial Z-stack images were acquired using FluoView software version 4.0 (Olympus, PA, USA) at an optical thickness of 1.5 µm. All images were acquired with identical settings for laser power, detector gain, and amplifier offset. A maximum Z-projection was performed in FIJI followed by rolling ball background subtraction to correct for uneven illumination. Image brightness and contrast were moderately adjusted using FIJI, with consistent adjustments made across images for each probe-fluorophore combination. ROIs were annotated on each image by overlaying the appropriate brain atlas image *Paxinos and Franklin, 2019*; *Bogovic et al., 2016* with guidance from DAPI staining and using the BigWarp plugin (https://imagej.net/BigWarp) in FIJI. These ROI outlines were used to generate binary masks in order to regionally restrict subsequent image analysis. Automated cell identification and analysis of fluorescent mRNA labeling was performed in CellProfiler (*McQuin et al., 2018*) using a modified version of the 'Colocalization' template pipeline (https://cellprofiler.org/examples). The pipeline was optimized to identify DAPI labeling (20–40 pixels

in diameter), in order to define cell outlines. This was followed by identification of fluorescent mRNA puncta (2–10 pixels in diameter) for each probe. Puncta that were localized within DAPI-identified cells (classified using the 'relate objects' module) were assigned to that cell for subsequent analysis. Quantification and further analysis/data visualization was performed using a custom-written Python code. Violin plots were made using the violin plot function in the Seaborn library *Waskom et al., 2020* of Python (with smoothing set to 0.2), and colocalization matrices were generated using the Seaborn heatmap function.

## In vivo microendoscopic calcium imaging

DAT::Cre mice received an injection of 300 nL AAV$_9$-Syn-FLEX-ChrimsonR-TdTomato in the DRN (ML:1.20, AP:–4.10, DV:–2.90; needle at a 20° angle from the midline, bevel facing medial), and 250 nL of AAV$_1$-Syn-GCaMP7f in the CeA (ML:2.85, AP:–1.20, DV:–4.75, needle bevel facing posterior). After ~4 weeks, mice underwent a second surgery to implant an integrated 0.6 mm diameter, 7.3 mm long gradient refractive index (GRIN) lens with attached baseplate (Inscopix, CA, USA) over the CeA (ML: 2.85, AP:–1.50, DV:–4.60). The lens was lowered slowly into the cleaned craniotomy by hand. The GRIN lens was adhered to the skull by a layer of adhesive cement (C&B Metabond; Parkell Inc, NY, USA) followed by a layer of black cranioplastic cement (Ortho-Jet; Lang, IL, USA), and protected by a small PCR tube cap, held in place by cement. The nVoke miniaturized microscope (Inscopix, CA, USA) consists of a 455±8 nm blue LED for GCaMP excitation, and a 620±30 nm red LED for simultaneous optogenetic manipulation (*Stamatakis et al., 2018*).

Behavioral experimentation commenced at least 1 week after baseplate surgery. Mice were first habituated to handling and connection of the microscope for a minimum of 3 consecutive days. For recording, mice were connected to the nVoke miniature microscope by tightening a small set screw on the baseplate. The microscope data cable was connected to a commutator (Inscopix, CA, USA), to allow unrestricted movement, and the commutator was itself connected to a data acquisition (DAQ) box. Grayscale images were acquired at a rate of 20 frames/s (fps; ~50ms exposure time) with the blue LED delivering 0.2–0.3 mW light power and analog gain on the image sensor set to 2. For the social approach task, mice were placed in the three-chamber apparatus (57.5lx22.5 w x 16.5h chamber with clear walls and grey floors). Following microscope connection, mice freely explored the chamber for 5 min. They were then confined to the center portion of the chamber, by the insertion of clear Plexiglas panels, during which a novel juvenile mouse was placed under one cup and a novel object was placed under the other cup. The panels were removed, and the test mouse was allowed to freely explore for a further 10 min. One 'group-housed' session was conducted without red-light delivery, and another 'group-housed' session was conducted with red 620 nm light delivery (8 pulses with 5ms pulse width, at 30 Hz, every 1 s; 10 mW) through the objective lens of the microscope, to activate ChrimsonR-expressing DRN[DAT] terminals. The order of 'group-housed' sessions was counterbalanced. 2–3 weeks later, mice were isolated for 24 hours, and another session ('isolated') commenced without red-light delivery. A top-down SLEAP (*Pereira et al., 2022*) (v1.3.1) model was trained using 774 labeled frames, annotating a skeleton composed of 15 keypoints (comprised of (1) nose, (2) head, (3) left ear, (4) right ear, (5) neck, (6) left forelimb, (7) right forelimb, (8) trunk, (9) left hindlimb, (10) right hindlimb, (11) tail base, (12-14) points along the length of the tail, and (15) tail tip). Pose estimation using the trained model was then performed on the behavior videos to determine if there was a social or object cup interaction. To determine if there was a social or object cup interaction, the nose of the mouse must be within 1.3 x the diameter of the cup, and the cup must be within a 90° cone in front of the mouse's head.

Raw videos of GCaMP fluorescence were first pre-processed in Inscopix Data Processing Software 1.3.0 (Inscopix, CA, USA) by cropping the region outside the GRIN lens, applying 2 x spatial downsampling, and a 3x3 median filter to fix defective pixels. A spatial band-pass filter was applied (0.005–0.5 oscillations/pixel) to remove high and low spatial frequency content, and rigid motion correction was performed (to account for small lateral displacements) by registering to a stable reference frame with a prominent landmark (e.g. blood vessel). Processed recordings were then exported as TIFF stacks for additional piecewise non-rigid motion correction using the NoRMCorre algorithm *Pnevmatikakis and Giovannucci, 2017* in overlapping 64x64 pixel grids using a MATLAB implementation. Constrained non-negative matrix factorization for endoscopic recordings (CNMF-E) was then used to extract the spatial shapes and calcium signals from individual cells in the imaging field of view

(*Zhou et al., 2018a*) using a MATLAB implementation (key parameters: minimum local correlation for seeding pixels = 0.9, minimum peak-to-noise ratio for seeding pixels = 12). The extracted calcium signals were inspected, and non-neuronal objects were manually excluded. All following downstream analyses used raw CNMF-E traces.

Calcium traces were aligned to detected behavioral events (interaction to the social and object cups, as determined by feature thresholds extracted from SLEAP keypoints). Single cell responses to social and object cup interaction were determined using an ROC (receiver operating characteristic) analysis, which has been previously described to determine neural responses to social behavior (*Kingsbury et al., 2019*; *Li et al., 2017*). A binary behavior vector of social or object cup interaction (calculated in 40ms time bins) was compared to a binary neural activity vectors generated by applying thresholds that span 100 steps from the minimum to maximum z-score value of each calcium trace to determine a true positive rate (TPR) and false positive rate (FPR) at each step. From these values, we yielded an ROC curve for each neuron that corresponded to the performance of that single neuron in predicting social or object cup interactions. The area under the ROC curve (auROC) was used to determine how strongly modulated each neuron was to the social and object stimuli. To determine the significance of single-cell responses, a null distribution of 1,000 auROC values was generated by randomly circularly shifting the binary behavior vectors and again comparing it to the binary neural activity signal. A neuron was considered to have a significant excitatory response to the social or object stimulus if the auROC value exceeded the 97.5th percentile of the 1000 shuffled auROC values and was considered to have a significant inhibitory response if the auROC value was less than the 2.5th percentile of the 1000 shuffled auROC values.

Co-registration of active neurons during imaging sessions was performed using CellReg (*Sheintuch et al., 2017*). In short, the spatial footprint matrices from each imaging session (as determined by CNMF-E) were used to align different imaging sessions within each animal to a reference session through translational and rotational shifts. Spatial correlation and centroid distance between cells were used to probabilistically register active cells across sessions.

Agglomerative hierarchical clustering was performed by averaging each neuron's response to the onset of social or object cup interaction throughout the trial. A social or object cup interaction was classified as a trial if it (a) lasted a minimum of 1 s, (b) if there had been at least 5 s that elapsed since the last interaction, and (c) if there was less than 1.5 s pause in interaction with the social or object cups. The z-scored averaged traces (5 s before and after the onset of social or object cup interaction) were concatenated, such that each row corresponds to one neuronal unit. Agglomerative hierarchical clustering was performed using MATLAB's 'cluster' function. Each neuron was initially designated as an individual cluster. Those that were in closest proximity were merged to form a new cluster, then the next closest were merged, etc. until a hierarchical tree was formed with all neurons contained within a single cluster. A threshold at 0.770×max(linkage) was set to prune branches from the hierarchical tree, so that all neurons below each cut were assigned to a single cluster. After the dendrogram was constructed, the average traces were displayed as a heatmap alongside their corresponding leaf. The traces of all neurons belonging to a single cluster were then averaged, and the number of neurons that corresponded to each behavior group was calculated for each cluster.

## Statistical analyses

Statistical tests were performed using GraphPad Prism 8 (GraphPad Software, CA, USA). Normality was evaluated using the D'Agostino-Pearson test, and data are expressed as mean ± standard error of the mean (SEM), unless otherwise noted. Data that followed a Gaussian distribution were compared using a paired or unpaired t-test (non-directional) for two experimental groups, and a one-way or two-way ANOVA with repeated measures for three or more experimental groups. Data for two experimental groups that did not follow a Gaussian distribution were compared using a Mann-Whitney $U$ test. Correlation between two variables was assessed using the Pearson's product-moment correlation coefficient. Threshold for significance was set at *$p < 0.05$, **$p < 0.01$, and ***$p < 0.001$.

## Acknowledgements

KMT is an HHMI Investigator, member of the Kavli Institute for Brain and Mind, the Wylie Vale Chair at the Salk Institute for Biological Studies, a New York Stem Cell Foundation - Robertson Investigator, and a McKnight Scholar. This work was supported by funding from the JPB Foundation, Alfred P Sloan

Foundation, New York Stem Cell Foundation, Klingenstein Foundation, McKnight Foundation, Clayton Foundation, Dolby Family Fund, R01-MH115920 (NIMH), R37-MH102441 (NIMH), the NIH Director's New Innovator Award DP2-DK102256 (NIDDK), and Pioneer Award DP1-AT009925 (NCCIH). GAM was supported by a Postdoctoral Research Fellowship from the Charles A King Trust. RLM was funded through the MSRP program in the Brains & Cognitive Sciences Department at MIT, supported by the Center for Brains, Minds and Machines (CBMM), and funded by NSF STC award CCF-1231216. EMW was supported by a summer scholarship from Johnson & Johnson. We thank C Leppla, J Olsen, P Namburi, V Barth, J Wang, K Batra, A Brown, and A Libster for technical advice, all members of the Tye Lab for helpful discussion, and advice from the CellProfiler team at the Broad Institute. We also thank Rachel Neve for the HSV construct, and Charu Ramakrishnan & Karl Deisseroth for AAV$_5$-fDIO-eYFP. This article is subject to HHMI's Open Access to Publications policy. HHMI lab heads have previously granted a nonexclusive CC BY 4.0 license to the public and a sublicensable license to HHMI in their research articles. Pursuant to those licenses, the author-accepted manuscript of this article can be made freely available under a CC BY 4.0 license immediately upon publication.

## Additional information

### Funding

| Funder | Grant reference number | Author |
|---|---|---|
| Salk Institute for Biological Studies | | Christopher R Lee<br>Gillian A Matthews<br>Mackenzie E Lemieux<br>Elizabeth M Wasserlein<br>Matilde Borio<br>Raymundo L Miranda<br>Laurel R Keyes<br>Gates P Schneider<br>Caroline Jia<br>Andrea Tran<br>Faith Aloboudi<br>May G Chan<br>Enzo Peroni<br>Grace Pereira<br>Alba López-Moraga<br>Anna Pallé<br>Eyal Y Kimchi<br>Nancy Padilla-Coreano<br>Romy Wichmann<br>Kay M Tye |
| Howard Hughes Medical Institute | | Christopher R Lee<br>Gillian A Matthews<br>Mackenzie E Lemieux<br>Elizabeth M Wasserlein<br>Matilde Borio<br>Raymundo L Miranda<br>Laurel R Keyes<br>Gates P Schneider<br>Caroline Jia<br>Andrea Tran<br>Faith Aloboudi<br>May G Chan<br>Enzo Peroni<br>Grace Pereira<br>Alba López-Moraga<br>Anna Pallé<br>Eyal Y Kimchi<br>Nancy Padilla-Coreano<br>Romy Wichmann<br>Kay M Tye |

| Funder | Grant reference number | Author |
|---|---|---|
| Kavli Foundation | | Christopher R Lee<br>Gillian A Matthews<br>Mackenzie E Lemieux<br>Elizabeth M Wasserlein<br>Matilde Borio<br>Raymundo L Miranda<br>Laurel R Keyes<br>Gates P Schneider<br>Caroline Jia<br>Andrea Tran<br>Faith Aloboudi<br>May G Chan<br>Enzo Peroni<br>Grace Pereira<br>Alba López-Moraga<br>Anna Pallé<br>Eyal Y Kimchi<br>Nancy Padilla-Coreano<br>Romy Wichmann<br>Kay M Tye |
| Dolby Family Ventures | | Christopher R Lee<br>Gillian A Matthews<br>Mackenzie E Lemieux<br>Elizabeth M Wasserlein<br>Matilde Borio<br>Raymundo L Miranda<br>Laurel R Keyes<br>Gates P Schneider<br>Caroline Jia<br>Andrea Tran<br>Faith Aloboudi<br>May G Chan<br>Enzo Peroni<br>Grace Pereira<br>Alba López-Moraga<br>Anna Pallé<br>Eyal Y Kimchi<br>Nancy Padilla-Coreano<br>Romy Wichmann<br>Kay M Tye |
| National Institutes of Health | R01-MH115920 (NIMH) | Christopher R Lee<br>Gillian A Matthews<br>Mackenzie E Lemieux<br>Elizabeth M Wasserlein<br>Matilde Borio<br>Raymundo L Miranda<br>Laurel R Keyes<br>Gates P Schneider<br>Caroline Jia<br>Andrea Tran<br>Faith Aloboudi<br>May G Chan<br>Enzo Peroni<br>Grace Pereira<br>Alba López-Moraga<br>Anna Pallé<br>Eyal Y Kimchi<br>Nancy Padilla-Coreano<br>Romy Wichmann<br>Kay M Tye |

| Funder | Grant reference number | Author |
|---|---|---|
| National Institutes of Health | R37-MH102441 (NIMH) | Christopher R Lee<br>Gillian A Matthews<br>Mackenzie E Lemieux<br>Elizabeth M Wasserlein<br>Matilde Borio<br>Raymundo L Miranda<br>Laurel R Keyes<br>Gates P Schneider<br>Caroline Jia<br>Andrea Tran<br>Faith Aloboudi<br>May G Chan<br>Enzo Peroni<br>Grace Pereira<br>Alba López-Moraga<br>Anna Pallé<br>Eyal Y Kimchi<br>Nancy Padilla-Coreano<br>Romy Wichmann<br>Kay M Tye |
| National Institutes of Health | DP1-AT009925 (NCCIH) | Christopher R Lee<br>Gillian A Matthews<br>Mackenzie E Lemieux<br>Elizabeth M Wasserlein<br>Matilde Borio<br>Raymundo L Miranda<br>Laurel R Keyes<br>Gates P Schneider<br>Caroline Jia<br>Andrea Tran<br>Faith Aloboudi<br>May G Chan<br>Enzo Peroni<br>Grace Pereira<br>Alba López-Moraga<br>Anna Pallé<br>Eyal Y Kimchi<br>Nancy Padilla-Coreano<br>Romy Wichmann<br>Kay M Tye |
| Clayton Foundation | | Christopher R Lee<br>Gillian A Matthews<br>Mackenzie E Lemieux<br>Elizabeth M Wasserlein<br>Matilde Borio<br>Raymundo L Miranda<br>Laurel R Keyes<br>Gates P Schneider<br>Caroline Jia<br>Andrea Tran<br>Faith Aloboudi<br>May G Chan<br>Enzo Peroni<br>Grace Pereira<br>Alba López-Moraga<br>Anna Pallé<br>Eyal Y Kimchi<br>Nancy Padilla-Coreano<br>Romy Wichmann<br>Kay M Tye |
| JPB Foundation | | Gillian A Matthews |
| Alfred P. Sloan Foundation | | Gillian A Matthews |

| Funder | Grant reference number | Author |
|---|---|---|
| New York Stem Cell Foundation | | Christopher R Lee<br>Gillian A Matthews<br>Mackenzie E Lemieux<br>Elizabeth M Wasserlein<br>Matilde Borio<br>Raymundo L Miranda<br>Laurel R Keyes<br>Gates P Schneider<br>Caroline Jia<br>Andrea Tran<br>Faith Aloboudi<br>May G Chan<br>Enzo Peroni<br>Grace Pereira<br>Alba López-Moraga<br>Anna Pallé<br>Eyal Y Kimchi<br>Nancy Padilla-Coreano<br>Romy Wichmann<br>Kay M Tye |
| Klingenstein Third Generation Foundation | | Christopher R Lee<br>Gillian A Matthews<br>Mackenzie E Lemieux<br>Elizabeth M Wasserlein<br>Matilde Borio<br>Raymundo L Miranda<br>Laurel R Keyes<br>Gates P Schneider<br>Caroline Jia<br>Andrea Tran<br>Faith Aloboudi<br>May G Chan<br>Enzo Peroni<br>Grace Pereira<br>Alba López-Moraga<br>Anna Pallé<br>Eyal Y Kimchi<br>Nancy Padilla-Coreano<br>Romy Wichmann<br>Kay M Tye |
| McKnight Foundation | | Christopher R Lee<br>Gillian A Matthews<br>Mackenzie E Lemieux<br>Elizabeth M Wasserlein<br>Matilde Borio<br>Raymundo L Miranda<br>Laurel R Keyes<br>Gates P Schneider<br>Caroline Jia<br>Andrea Tran<br>Faith Aloboudi<br>May G Chan<br>Enzo Peroni<br>Grace Pereira<br>Alba López-Moraga<br>Anna Pallé<br>Eyal Y Kimchi<br>Nancy Padilla-Coreano<br>Romy Wichmann<br>Kay M Tye |

| Funder | Grant reference number | Author |
|---|---|---|
| National Institutes of Health | DP2-DK102256 (NIDDK) | Christopher R Lee<br>Gillian A Matthews<br>Mackenzie E Lemieux<br>Elizabeth M Wasserlein<br>Matilde Borio<br>Raymundo L Miranda<br>Laurel R Keyes<br>Gates P Schneider<br>Caroline Jia<br>Andrea Tran<br>Faith Aloboudi<br>May G Chan<br>Enzo Peroni<br>Grace Pereira<br>Alba López-Moraga<br>Anna Pallé<br>Eyal Y Kimchi<br>Nancy Padilla-Coreano<br>Romy Wichmann<br>Kay M Tye |
| Charles A. King Trust | Postdoctoral Research Fellowship | Gillian A Matthews |
| Massachusetts Institute of Technology | Brains & Cognitive Sciences Department | Raymundo L Miranda |
| Massachusetts Institute of Technology | | Raymundo L Miranda |
| National Science Foundation | CCF-1231216 | Raymundo L Miranda |
| Johnson and Johnson | Summer Scholarship | Elizabeth M Wasserlein |

The funders had no role in study design, data collection and interpretation, or the decision to submit the work for publication.

## Author contributions

Christopher R Lee, Conceptualization, Resources, Data curation, Software, Formal analysis, Supervision, Validation, Investigation, Visualization, Methodology, Writing – original draft, Project administration, Writing – review and editing, Performed stereotaxic surgeries, Ran optogenetic manipulation experiments and analyzed behavioral data, Performed immunohistochemistry and analyzed images, Performed in vivo calcium imaging and analysis, Reviewed, organized and prepared the data for data sharing; Gillian A Matthews, Conceptualization, Resources, Data curation, Software, Formal analysis, Supervision, Validation, Investigation, Visualization, Methodology, Writing – original draft, Project administration, Writing – review and editing, Performed stereotaxic surgeries, Ran optogenetic manipulation experiments and analyzed behavioral data, Performed immunohistochemistry and analyzed images, Performed ex vivo electrophysiology, Reviewed, organized and prepared the data for data sharing; Mackenzie E Lemieux, Formal analysis, Investigation, Writing – review and editing, Performed and analyzed smFISH experiments, Performed stereotaxic surgeries, Ran optogenetic manipulation experiments and analyzed behavioral data; Elizabeth M Wasserlein, Formal analysis, Investigation, Writing – review and editing, Performed stereotaxic surgeries, Ran optogenetic manipulation experiments and analyzed behavioral data, Performed immunohistochemistry and analyzed images; Matilde Borio, Formal analysis, Investigation, Writing – review and editing, Performed and analyzed smFISH experiments, Ran optogenetic manipulation experiments and analyzed behavioral data; Raymundo L Miranda, Investigation, Writing – review and editing, Ran optogenetic manipulation experiments and analyzed behavioral data; Laurel R Keyes, Data curation, Formal analysis, Writing – review and editing, Performed Markov model analysis, Reviewed, organized and prepared the data for data sharing; Gates P Schneider, Formal analysis, Investigation, Writing – review and editing, Performed immunohistochemistry and analyzed images; Caroline Jia, Formal analysis, Investigation, Writing – review and editing, Performed stereotaxic surgeries, Ran optogenetic manipulation experiments and analyzed behavioral data; Andrea Tran, Investigation, Writing – review and editing, Ran optogenetic manipulation experiments and analyzed behavioral data; Faith Aloboudi, Investigation, Writing – review

and editing, Ran optogenetic manipulation experiments and analyzed behavioral data; May G Chan, Investigation, Writing – review and editing, Performed immunohistochemistry and analyzed images; Enzo Peroni, Investigation, Writing – review and editing, Performed stereotaxic surgeries, Ran optogenetic manipulation experiments and analyzed behavioral data, Performed immunohistochemistry and analyzed images; Grace Pereira, Investigation, Writing – review and editing, Performed stereotaxic surgeries, Ran optogenetic manipulation experiments and analyzed behavioral data, Performed immunohistochemistry and analyzed images; Alba López-Moraga, Investigation, Writing – review and editing, Performed stereotaxic surgeries, Ran optogenetic manipulation experiments and analyzed behavioral data, Performed immunohistochemistry and analyzed images; Anna Pallé, Investigation, Writing – review and editing, Performed stereotaxic surgeries, Ran optogenetic manipulation experiments and analyzed behavioral data, Performed immunohistochemistry and analyzed images; Eyal Y Kimchi, Writing – review and editing, Contributed to experimental design and data interpretation; Nancy Padilla-Coreano, Writing – review and editing, Contributed to experimental design and data interpretation; Romy Wichmann, Data curation, Writing – review and editing, Contributed to experimental design and data interpretation, Reviewed, organized and prepared the data for data sharing; Kay M Tye, Conceptualization, Supervision, Funding acquisition, Writing – original draft, Writing – review and editing

**Author ORCIDs**
Christopher R Lee ⑩ https://orcid.org/0000-0002-5952-9924
Gillian A Matthews ⑩ https://orcid.org/0000-0001-6754-0333
Mackenzie E Lemieux ⑩ https://orcid.org/0000-0001-6015-8668
Raymundo L Miranda ⑩ https://orcid.org/0009-0009-9694-1545
Laurel R Keyes ⑩ https://orcid.org/0000-0001-5433-9948
May G Chan ⑩ https://orcid.org/0009-0005-6664-3521
Grace Pereira ⑩ https://orcid.org/0000-0003-4371-7020
Alba López-Moraga ⑩ https://orcid.org/0000-0002-2084-4855
Eyal Y Kimchi ⑩ https://orcid.org/0000-0003-4327-1102
Nancy Padilla-Coreano ⑩ https://orcid.org/0000-0001-9293-2697
Romy Wichmann ⑩ https://orcid.org/0000-0002-4506-8813
Kay M Tye ⑩ https://orcid.org/0000-0002-2435-0182

**Ethics**
All procedures involving animals were conducted in accordance with NIH guidelines and approved by the MIT Committee on Animal Care or the Salk Institute Institutional Animal Care and Use Committee (IACUC protocol #18-00060). Mice (>7 weeks of age) were anaesthetized with isoflurane (inhalation: 4% for induction, ~2% for maintenance, oxygen flow rate 1 L/min) before being placed in a digital small animal stereotax. Surgeries were performed under aseptic conditions with body temperature maintained by a heating pad throughout, and every effort was made to minimize suffering.

Reviewer #1 (Public review): https://doi.org/10.7554/eLife.105955.3.sa1
Reviewer #2 (Public review): https://doi.org/10.7554/eLife.105955.3.sa2
Reviewer #3 (Public review): https://doi.org/10.7554/eLife.105955.3.sa3
Author response https://doi.org/10.7554/eLife.105955.3.sa4

---

# Additional files

**Supplementary files**
MDAR checklist

**Data availability**
Source data files have been provided for all figures. Ex vivo electrophysiology and in vivo calcium imaging data have been deposited to DANDI Archive.

The following dataset was generated:

| Author(s) | Year | Dataset title | Dataset URL | Database and Identifier |
|---|---|---|---|---|
| Keyes L, Lee CR, Wichmann R, Matthews GA, Kay M | 2025 | Separable dorsal raphe dopamine projections mediate the facets of loneliness-like state | https://doi.org/10.48324/dandi.001195/0.250408.1733 | DANDI Archive, 10.48324/dandi.001195/0.250408.1733 |

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
