## [Editor Report · eLife Assessment]

This study dissects the function of 3 outputs of a specific population of modulatory neurons, dorsal raphe dopamine neurons, in social and affective behavior. It provides **valuable** information that both confirms prior results and provides new insights. The strength of the evidence is **convincing**, based on cutting-edge approaches and analysis. This study will be of interest to behavioral and systems neuroscientists, especially those interested in social and emotional behavior.

---

## [Referee Report · Reviewer #1 (Public review)]

Summary:

The authors had previously found that a brief social isolation could increase the activity of these neurons, and that manipulation of these neurons could alter social behavior in a social rank dependent fashion. This manuscript explored which of the outputs were responsible for this, identifying the central nucleus of the amygdala as the key output region. The authors identified some discrete behavior changes associated with these outputs, and found that during photostimulation of these outputs, neuronal activity appeared altered in 'social response' neurons. In the revised manuscript, the authors address the comments in a rigorous fashion.

Strengths:

Rigorous analysis of the anatomy. Careful examination of the hetergenous effects on cell activity due to stimulation, linking the physiology with the behavior via photostimulation during recording in vivo.

Weaknesses:

The authors have responded to all of my comments.

---

## [Referee Report · Reviewer #2 (Public review)]

Summary:

The authors perform a series of studies to follow up on their previous work, which established a role for dorsal raphe dopamine neurons (DRN) in the regulation of social-isolation-induced rebound in mice. In the present study, Lee et. al, use a combination of modern circuit tools to investigate putatively distinct roles of DRN dopamine transporting containing (DAT) projections to the bed nucleus of the stria terminalis (BNST), central amygdala (CeA), and posterior basolateral amygdala (BLP). Notably, they reveal that optogenetic stimulation of distinct pathways confers specific behavioral states, with DRNDAT-BLP driving aversion, DRNDAT-BNST regulating non-social exploratory behavior, and DRNDAT-CeA promoting socialability. A combination of electrophysiological studies and in situ hybridization studies reveal heterogenous dopamine and neuropeptide expression and different firing properties, providing further evidence of pathway-specific neural properties. Lastly, the authors combine optogenetics and calcium imaging to resolve social encoding properties in the DRNDAT-CeA pathway, which correlates observed social behavior to socially engaged neural ensembles.

Collectively, these studies provide an interesting way of dissecting out separable features of a complex multifaceted social-emotional state that accompanies social isolation and the perception of 'loneliness.' The main conclusions of the paper provide an important and interesting set of findings that increase our understanding of these distinct DRN projections and their role in a range of social (e.g., prosocial, dominance), non-social, and emotional behaviors. However, as noted below, the examination of these circuits within a homeostatic framework is limited given that a number of the datasets did not include an isolated condition. The DRNDAT-CeA pathway was investigated with respect to social homeostatic states in the present study for some of the datasets.

Strengths:

(1) The authors perform a comprehensive and elegant dissection of the anatomical, behavioral, molecular, and physiological properties of distinct DRN projections relevant to social, non-social, and emotional behavior, to address multifaceted and complex features of social state.

(2) This work builds on prior findings of isolation-induced changes in DRN neurons and provides a working framework for broader circuit elements that can be addressed across social homeostatic state.

(3) This work characterizes a broader circuit implicated in social isolation and provides a number of downstream targets to explore, setting a nice foundation for future investigation.

(4) The studies account for social rank and anxiety-like behavior in several of the datasets, which are important consideration to the interpretation of social motivation states, especially in male mice with respect to dominance behavior.

Weaknesses:

(1) The conceptual framework of the study is based on the premise of social isolation and perceived 'loneliness' under the framework of social homeostasis, analogous to hunger. In this framework, social isolation should provoke an aversive state and compensatory social contact behavior. In the authors' prior work, they demonstrate synaptic changes in DRN neurons and social rebound following acute social isolation. Thus, the prediction would be that downstream projections also would show state dependent changes as a function of social isolation state (e.g., grouped/socially engaged vs. isolated). In the current paper, a social isolation condition was included for some but not all experiments, which should be considered in the interpretation of the data, specifically within the context of dynamic isolation states.

(2) Figure 1 confirms co-laterals in the BNST and CeA via anatomical tracing studies. The goal of the optogenetic studies is to dissociate functional/behavioral roles of distinct projections. One limitation of optogenetic projection targeting is the possibility of back-propagating action potentials (stimulation of terminals in one region may back-propagate to activate cell bodies, and then afferent projections to other regions), and/or stimulation of fibers of passage. However, this is addressed in the discussion and the present data are convincing, which minimizes the concern.

(3) Sex as a biological variable should be considered in the present data, as included in the discussion.

---

## [Referee Report · Reviewer #3 (Public review)]

Summary:

The authors investigated the role of dopaminergic neurons (dopamine transporter expressing, DAT) in the dorsal raphe nucleus (DRN) in regulating social and affective behavior through projections to the central nucleus of the amygdala (CeA), bed nucleus of the stria terminalis (BNST), and the posterior subdivision of the basolateral amygdala. The largest effect observed was in the DRN-DAT projections to the CeA. Augmenting previously published results from this group (Matthews et al., 2016), the comprehensive behavioral analysis relative to social dominance, gene expression analysis, electrophysiological profiling, and in vivo imaging provides novel insights into how DRN-DAT projections to the CeA influence the engagement of social behavior in the contexts of group housed and socially isolated mice.

Strengths:

Correlational analysis with social dominance is a nice addition to the study. The overall computational analyses performed are well-designed and rigorous.

Weaknesses:

(1) Analysis of dopamine receptor expression did not include Drd3, Drd4, or Drd5 which may provide more insights into how dopamine modulates downstream targets. This is particularly relevant to the BNST projection in which the densest innervation did not robustly co-localize with the expression of either Drd1 or Drd2. It is also possible that dopamine release from DRN-DAT neurons in any or all of these structures in modulating neurotransmitter release from inputs to these regions that contain D2 receptors on their terminals.

(2) Although not the focus of this study, without pharmacological blockade of dopamine receptors, it is not possible to assess what the contribution of dopamine is to the behavioral outcomes. Given the co-release of glutamate and GABA from these neurons it is possible that dopamine plays only a marginal role in the functional connectivity of DRN-DAT neurons.

(3) Photostimulation parameters used during the behavioral studies (8 pulses of light delivered at 30 Hz for several minutes) could lead to confounding results limiting data interpretation. As shown in Figure 6J, 8 pulses of light delivered at 30 Hz results in a significant attenuation of the EPSC amplitude in the BLP and CeA projection. Thus, prolonged stimulation could lead to significant synaptic rundown resulting in an overall suppression of connectivity in the later stages of the behavioral analyses.

Comments on revisions:

No further issues have been identified.

---

## [Author Response]

The following is the authors’ response to the original reviews

**Reviewer #1 (Public review):**
Summary:The authors had previously found that brief social isolation could increase the activity of these neurons, and that manipulation of these neurons could alter social behavior in a social rank-dependent fashion. This manuscript explored which of the outputs were responsible for this, identifying the central nucleus of the amygdala as the key output region. The authors identified some discrete behavior changes associated with these outputs, and found that during photostimulation of these outputs, neuronal activity appeared altered in 'social response' neurons.Strengths:Rigorous analysis of the anatomy. Careful examination of the heterogenous effects on cell activity due to stimulation, linking the physiology with the behavior via photostimulation during recording in vivo.Weaknesses:(1) There are some clear imbalances in the sample size across the different regions parsed. The CeA has a larger sample size, likely in part to the previous work suggesting differential effects depending on social rank/dominance. Given the potential variance, it may be hard to draw conclusions about the impact of stimulation across different social ranks for other groups.

While it may be difficult to draw conclusions about the impact of stimulation across different social ranks, we believe that the dominance-induced variance in our dataset reveals key insights into how social history may affect the function of these circuits. However, we do recognize that there are imbalances in sample size across the different circuits that we probed. To test whether we could detect a significant effect in our DRN^DAT^-CeA:ChR2 group with a sample size matched to the DRN^DAT^-BLP:ChR2 group (the lowest sample size of the three circuits probed), we subsampled and ran tests for statistical significance using the following MATLAB code:

**Author response image 1. sa4fig1:** 

We found that out of 1000 subsamples, we detected a statistically significant effect 40.5% of the time (Author response image 2A). This suggests that the optogenetic effect exists, though it is moderate and is variable across mice (as explained by the significant correlation between social rank and optogenetic effect).

To test whether these inconsistent effects may be an effect of variance induced by social rank, we wrote the following MATLAB code to maintain the distribution of social rank in our subsamples:

**Author response image 2. sa4fig2:** P-values from subsampling analysis show a moderately reproducible social preference effect in DRN^DAT^-CeA:ChR2 mice, but not in DRN^DAT^-BNST:ChR2 mice. (A-D) Histograms showing distribution of paired t-test p-values comparing OFF and ON social preference scores (as shown in Figure 4A-I) in subsampled groups (to match the sample size of the DRN^DAT^-BLP:ChR2 group). (A) 14 DRN^DAT^-CeA:ChR2 mice were randomly subsampled, a paired t-test was performed, and the resulting p-values were binned and plotted. (B) Same as (A), but ensuring that the proportion of subordinate, intermediate, and dominant mice in the subsampled groups were the same as the original distribution. (C) Same as (A), but with DRN^DAT^-BNST:ChR2 mice. (D) Same as (B), but with DRN^DAT^-BNST:ChR2 mice.

**Author response image 3. sa4fig3:** 

We found that out of 1000 subsamples, we detected a statistically significant effect 45.5% of the time when we maintained the original distribution of social rank in DRN^DAT^-CeA:ChR2 mice (Author response image 2B). This suggests that reducing the sample size to N=14 reduces the statistical power and indeed can make an effect harder to reliably detect. The reviewer is correct in saying that sample imbalance may skew conclusions. However, given the rank-dependent optogenetic effect on social preference seen in DRN^DAT^-CeA:ChR2 mice (N=29 mice, p=0.002, Figure 4H) that is notably absent in DRN^DAT^-BLP:ChR2 mice (N=14 mice, p=0.806, Figure 4I), we hypothesize that we would not see a significant effect of photoactivating the DRN^DAT^-BLP circuit on social preference, even with a larger sample size. While we acknowledge there may be evidence that there could be an effect in the DRN^DAT^-BLP projection, this analysis reveals that this effect is not as robust as the effect we see in the DRN^DAT^-CeA projection, which is the focus of this study. An in-depth exploration of the DRN^DAT^ projection to the BLP is certainly warranted in future studies.

Interestingly, the same analysis approach applied to DRN^DAT^-BNST:ChR2 mice suggest a reliably negative result, with subsampling only resulting in a significant result 1.1% of the time (Author response image 2C) and 1.7% of the time if maintaining the original rank distribution (Author response image 2D).

(2) It is somewhat unclear why only the 'social object ratio' was used to assess the effects versus more direct measurements of social behavior.

We decided to use ‘social:object ratio’ as we felt that measurement more directly supported our claim of increased social preference through optogenetic manipulation; however, in our updated manuscript, we included direct measurements of social behavior in the revised manuscript (Figure 4—figure supplement 1) and have updated the legend to reflect this addition (lines 1679-1684; 1698-1708).

(3) Somewhat related, while it is statistically significant, it is unclear if the change seen in face investigation of biologically significant, on average, it looks like a few-seconds difference and that was not modulated by social rank.

While the effect size is relatively small (4.19 seconds, 2.32% of the session), we believe we should report any statistically significant findings we discover. However, due to the small effect size, we have de-emphasized our claims regarding this finding in the text (line 172).

(4) There are several papers studying these neurons that have explored behaviors examined here, as well as the physiological connectivity that are not cited that would provide important context for this work. In particular, multiple groups have found a dopamine-mediated IPSP in the BNST, in contrast to this work. There are technical differences that may drive these differences, but not addressing them is a major weakness.

In the revised text, we have cited the groups who have found different effects of dopamine-mediated effects in the ovBNST (specifically from Krawczyk et al., 2011, Maracle et al., 2018, and Yu et al., 2021) and reconciled these results with those from our study (lines 422-432).

(5) The inclusion of some markers for receptors for some of these outputs is interesting, and the authors suggest that this may be important, but this is somewhat disconnected from the rest of the work performed.

We agree that we cannot make any causal signaling mechanism claims with the current downstream receptor RNA expression data (and we are careful in avoiding making those claims in the text), but we include these data to offer a potential mechanism and hope that these descriptive data will be useful to the field for follow up studies.

**Reviewer #2 (Public review):**
Summary:The authors perform a series of studies to follow up on their previous work, which established a role for dorsal raphe dopamine neurons (DRN) in the regulation of social-isolation-induced rebound in mice. In the present study, Lee et. al, use a combination of modern circuit tools to investigate putatively distinct roles of DRN dopamine transporting containing (DAT) projections to the bed nucleus of the stria terminalis (BNST), central amygdala (CeA), and posterior basolateral amygdala (BLP). Notably, they reveal that optogenetic stimulation of distinct pathways confers specific behavioral states, with DRNDAT-BLP driving aversion, DRNDAT-BNST regulating non-social exploratory behavior, and DRNDAT-CeA promoting socialability. A combination of electrophysiological studies and in situ hybridization studies reveal heterogenous dopamine and neuropeptide expression and different firing properties, providing further evidence of pathway-specific neural properties. Lastly, the authors combine optogenetics and calcium imaging to resolve social encoding properties in the DRNDAT-CeA pathway, which correlates observed social behavior to socially engaged neural ensembles.

Collectively, these studies provide an interesting way of dissecting out separable features of a complex multifaceted social-emotional state that accompanies social isolation and the perception of 'loneliness.' The main conclusions of the paper provide an important and interesting set of findings that increase our understanding of these distinct DRN projections and their role in a range of social (e.g., prosocial, dominance), non-social, and emotional behaviors. However, as noted below, the examination of these circuits within a homeostatic framework is limited given that a number of the datasets did not include an isolated condition. The DRNDAT-CeA pathway was investigated with respect to social homeostatic states in the present study for some of the datasets.

Strengths:(1) The authors perform a comprehensive and elegant dissection of the anatomical, behavioral, molecular, and physiological properties of distinct DRN projections relevant to social, non-social, and emotional behavior, to address multifaceted and complex features of social state.(2) This work builds on prior findings of isolation-induced changes in DRN neurons and provides a working framework for broader circuit elements that can be addressed across the social homeostatic state.(3) This work characterizes a broader circuit implicated in social isolation and provides a number of downstream targets to explore, setting a nice foundation for future investigation.(4) The studies account for social rank and anxiety-like behavior in several of the datasets, which are an important consideration to the interpretation of social motivation states, especially in male mice with respect to dominance behavior.Weaknesses:(1) The conceptual framework of the study is based on the premise of social isolation and perceived 'loneliness' under the framework of social homeostasis, analogous to hunger. In this framework, social isolation should provoke an aversive state and compensatory social contact behavior. In the authors' prior work, they demonstrate synaptic changes in DRN neurons and social rebound following acute social isolation. Thus, the prediction would be that downstream projections also would show state-dependent changes as a function of social housing conditions (e.g., grouped vs. isolated). In the current paper, a social isolation condition was not included for the majority of the studies conducted (e.g., Figures 1-6 do not include an isolated condition, Figures 7-8 do include an isolated condition). Thus, while Figure 1-6 adds a very interesting and compelling set of data that is of high value to the social behavior field with respect to social and emotional processing and general circuit characterization, these studies do not directly investigate the impacts of dynamic social homeostatic state. The main claim of the paper, including the title (e.g., separable DRN projections mediate facets of loneliness-like state), abstract, intro, and discussion presents the claim of this work under the framework of dynamic social homeostatic states, which should be interpreted with caution, as the majority of the work in the paper did not include a social isolation comparison.

In previous studies, loneliness-like phenotypes have been characterized across species as having the key dimensions of an aversive state that increases prosociality[1–5]. These two features are amplified by photostimulation of DRN DA neurons, and as we show in this manuscript, are separable across different projections to each target, and our ability to distinctly mimic different aspects of the constellation of features we characterize as “loneliness.”

However we agree with the reviewer that we do not intend to imply that the mouse currently feels lonely. Indeed, isolating the animals would occlude our ability to see photostimulation-induced mimicry of specific features of the loneliness-like phenotype, and this is precisely why we did not isolate animals for our ChR2 gain-of-function experiments. To address the reviewers’ concern, we will change the title of our manuscript from making a claim of “mediating” which we agree would rely more heavily on mediating actual (ethologically-induced) loneliness rather than “mimicry” (photostimulation-induced) behaviors associated with a loneliness-like phenotype. We have changed language regarding this claim throughout our manuscript (Lines 1, 83, 285, 369).

For the ChR2 experiments in particular, we intended the optogenetic manipulation to be a gain-of-function one to test the hypothesis that activation of these circuits is sufficient to recapitulate different facets of a loneliness-like state (i.e. prosociality, aversion, and increased exploratory behavior). As such, that is why we only included group-housed conditions for these experiments—to mimic the phenotype of social isolation without social isolation. To test the necessity of these circuits in mediating different facets of a loneliness-like state, we agree that silencing the studied projections in an isolated state is critical, which is what we show in Figure 8. We agree that the addition of an isolated condition to understand the circuit-specific impact of dynamic social homeostatic state is important (particularly through in vivo recordings of these specific circuits during relevant behaviors), and would be a great follow-up to this study.

(2) In Figure 1, the authors confirm co-laterals in the BNST and CeA via anatomical tracing studies. The goal of the optogenetic studies is to dissociate the functional/behavioral roles of distinct projections. However, one limitation of optogenetic projection targeting is the possibility of back-propagating action potentials (stimulation of terminals in one region may back-propagate to activate cell bodies, and then afferent projections to other regions), and/or stimulation of fibers of passage. Therefore, one limitation in the dataset for the optogenetic stimulation studies is the possibility of non-specific unintended activation of projections other than those intended (e.g., DRNDAT-CeA). This can be dealt with by administering lidocaine to prevent back-propagating action potentials.

While back-propagating action potentials are potentially confounding for the manipulation techniques presented in this paper, we do show circuit-specific optogenetic behavioral effects *despite* significant collateralization (specifically between DRN^DAT^ neurons projecting to the CeA and BNST; Figure 1H), suggesting circuit-specificity. Namely, we see that stimulation of DRN^DAT^ terminals in CeA promotes social preference (Figure 4E,K) whereas stimulation of DRN^DAT^ terminals in BNST promotes rearing (exploratory) behavior (Figure 3G). There is a non-negligible chance that we are stimulating DRN^DAT^ fibers of passage, which we have addressed in a caveat disclaimer included in the revised discussion (lines 345-347).

(3) It is unclear from the test, but in the subjects' section of the methods, it appears that only male animals were included in the study, with no mention of female subjects. It should be clear to the reader that this was conducted in males only if that is the case, with consideration or discussion, about female subjects and sex as a biological variable.

In the revised manuscript, we have included discussion about sex as a biological variable (lines 342-345).

(4) Averaged data are generally reported throughout the study in the form of bar graphs, across most figures. Individual data points would increase the transparency of the data.

In an effort to increase the transparency of the data, we have prepared source data for each data panel in the final version of the manuscript and will upload it to eLife.

REFERENCES

(1) Cacioppo, J.T., Hughes, M.E., Waite, L.J., Hawkley, L.C., and Thisted, R.A. (2006). Loneliness as a specific risk factor for depressive symptoms: cross-sectional and longitudinal analyses. Psychol Aging 21, 140–151. https://doi.org/10.1037/0882-7974.21.1.140.

(2) Cacioppo, S., Capitanio, J.P., and Cacioppo, J.T. (2014). Toward a Neurology of Loneliness. Psychol Bull 140, 1464–1504. https://doi.org/10.1037/a0037618.

(3) Baumeister, R.F., and Leary, M.R. (1995). The need to belong: Desire for interpersonal attachments as a fundamental human motivation. Psychological Bulletin 117, 497–529. https://doi.org/10.1037/0033-2909.117.3.497.

(4) Niesink, R.J., and Van Ree, J.M. (1982). Short-term isolation increases social interactions of male rats: A parametric analysis. Physiology & Behavior 29, 819–825. https://doi.org/10.1016/0031-9384(82)90331-6.

(5) Panksepp, J., and Beatty, W.W. (1980). Social deprivation and play in rats. Behavioral & Neural Biology 30, 197–206. https://doi.org/10.1016/S0163-1047(80)91077-8.

**Reviewer #3 (Public review):**
Summary:The authors investigated the role of dopaminergic neurons (dopamine transporter expressing, DAT) in the dorsal raphe nucleus (DRN) in regulating social and affective behavior through projections to the central nucleus of the amygdala (CeA), bed nucleus of the stria terminalis (BNST), and the posterior subdivision of the basolateral amygdala. The largest effect observed was in the DRN-DAT projections to the CeA. Augmenting previously published results from this group (Matthews et al., 2016), the comprehensive behavioral analysis relative to social dominance, gene expression analysis, electrophysiological profiling, and in vivo imaging provides novel insights into how DRN-DAT projections to the CeA influence the engagement of social behavior in the contexts of group-housed and socially isolated mice.Strengths:Correlational analysis with social dominance is a nice addition to the study. The overall computational analyses performed are well-designed and rigorous.Weaknesses:(1) Analysis of dopamine receptor expression did not include Drd3, Drd4, or Drd5 which may provide more insights into how dopamine modulates downstream targets. This is particularly relevant to the BNST projection in which the densest innervation did not robustly co-localize with the expression of either Drd1 or Drd2. It is also possible that dopamine release from DRN-DAT neurons in any or all of these structures modulates neurotransmitter release from inputs to these regions that contain D2 receptors on their terminals.

Although we find that there is more *Vipr2* and *Npbwr1* expression compared to *Drd1* and *Drd2* expression in ovBNST, we still do find that a substantial proportion of cells in ovBNST express dopamine receptors (particularly D2 dopamine receptors, as shown in Figure 5C). In our revised manuscript, we have discussed potential functional mechanism through D3, D4, and D5 dopamine receptors, as well as pre-synaptic dopamine receptor expression (lines 459-461).

(2) Although not the focus of this study, without pharmacological blockade of dopamine receptors, it is not possible to assess what the contribution of dopamine is to the behavioral outcomes. Given the co-release of glutamate and GABA from these neurons, it is possible that dopamine plays only a marginal role in the functional connectivity of DRN-DAT neurons.

While we agree with the reviewer’s comments, we are careful to avoid making claims about dopamine-mediated physiological and behavioral effects of DRN^DAT^ neurons (despite that these neurons are genetically identified through the expression of dopamine transporter [DAT]), mentioned in lines 222-228 in the text.

(3) Photostimulation parameters used during the behavioral studies (8 pulses of light delivered at 30 Hz for several minutes) could lead to confounding results limiting data interpretation. As shown in Figure 6J, 8 pulses of light delivered at 30 Hz result in a significant attenuation of the EPSC amplitude in the BLP and CeA projection. Thus, prolonged stimulation could lead to significant synaptic rundown resulting in an overall suppression of connectivity in the later stages of the behavioral analyses.

Despite attenuation of EPSC amplitude in BLP and CeA projections and potential synaptic rundown, we still observe significant behavioral effects through optogenetic manipulation of these circuits (increasing the likelihood of capturing a ‘true positive’ rather than a ‘false negative’ effect). In general, we attempt to reduce the duty cycle by sparingly delivering trains of optogenetic stimulation (eight 5-ms pulses every 5 seconds). Additionally, in the real time place preference task where stimulation of the DRN^DAT^-BLP projection significantly reduces the time spent in the “ON” chamber, stimulation is only delivered when the mouse is in the “ON” compartment of the apparatus. However, we do feel that the reviewer’s concern that EPSC attenuation and potential synaptic rundown may potentially explain the robust place avoidance effects in DRN^DAT^-BLP:ChR2 mice in the first half of the session (Figure 2G). Importantly, we show in our previous published work (Matthews et al., 2016, Cell; Figure 3) through fast-scan cyclic voltammetry (FSCV) that dopamine transients were consistently recorded in response to eight pulses of 30 Hz DRN^TH^ stimulation delivered every 5 seconds in the BNST, though less consistently in the CeA.